# HPK1 citron homology domain regulates phosphorylation of SLP76 and modulates kinase domain interaction dynamics

Avantika S. Chitre[1], Ping Wu[1], Benjamin T. Walters [1], Xiangdan Wang[1], Alexandre Bouyssou [1], Xiangnan Du[1], Isabelle Lehoux[1,2], Rina Fong[1], Alisa Arata[1], Joyce Chan[1], Die Wang [1], Yvonne Franke[1], Jane L. Grogan [1,3], Ira Mellman[1] ✉, Laetitia Comps-Agrar [1] ✉ & Weiru Wang[1,4] ✉

Hematopoietic progenitor kinase 1 (HPK1) is a negative regulator of T-cell receptor signaling and as such is an attractive target for cancer immunotherapy. Although the role of the HPK1 kinase domain (KD) has been extensively characterized, the function of its citron homology domain (CHD) remains elusive. Through a combination of structural, biochemical, and mechanistic studies, we characterize the structure-function of CHD in relationship to KD. Crystallography and hydrogen-deuterium exchange mass spectrometry reveal that CHD adopts a seven-bladed β-propellor fold that binds to KD. Mutagenesis associated with binding and functional studies show a direct correlation between domain-domain interaction and negative regulation of kinase activity. We further demonstrate that the CHD provides stability to HPK1 protein in cells as well as contributes to the docking of its substrate SLP76. Altogether, this study highlights the importance of the CHD in the direct and indirect regulation of HPK1 function.

The activation of T cells in response to antigen recognition is fundamental to the adaptive immune response and reflects an integration of signaling events generated by the T-cell receptor (TCR), costimulatory receptors, and receptors for various cytokines. TCR and costimulatory receptor signaling are carefully regulated so as to prevent excessive immune activation, T-cell depletion (via activation-induced cell death; AICD), T-cell exhaustion, and potential tissue damage. Costimulatory receptors, such as CD28 and CD226, are controlled by coinhibitory receptors (PD-1 and TIGIT, respectively)[1]. While coinhibitory receptors may also impact TCR signaling indirectly, a variety of cytosolic components are known that appear dedicated to this purpose. These include the E3 ubiquitin ligase Casitas B-lineage lymphoma proto-oncogene-b (Cbl-b)[2], the protein tyrosine phosphatase non-receptor type 22 (PTPN22) phosphatase[3], and the non-receptor hematopoietic progenitor kinase 1 (HPK1 or MAP4K1)[4]. Since taking steps to augment

T-cell function would be expected to increase anti-tumor immunity, there has been a resurgence of interest in these negative regulators, given that each represents a potential therapeutic target.

As a kinase, and therefore considered druggable, HPK1 has emerged of particular interest[4–8]. Upon antigen recognition by TCR, lymphocyte-specific protein tyrosine kinase (Lck) phosphorylates CD3 and subsequently phosphorylates and activates the zeta chain of T-cell receptor associated protein kinase 70 (Zap70). Zap70 in turn phosphorylates linker for activation of T cells (LAT), leading to docking of key downstream components of the pathway including the SH2 domain-containing leukocyte protein of 76 kD (SLP76). This results in signaling mediated by phospholipase C γ (PLCγ) and mitogen-activated protein kinase (MAPK) pathways, the latter including activation of extracellular signal-regulated kinase 1/2 (ERK) and c-Jun N-terminal kinase (JNK), essential to T-cell activation and cytokine

[1]Genentech, Inc., 1 DNA Way, South San Francisco, CA 94080, USA. [2]Present address: Gilead Sciences, Inc., 333 Lakeside Drive, Foster City, CA 94404, USA. [3]Present address: GraphiteBio, Incl., 1400 Sierra Point Parkway, Brisbane, CA 94005, USA. [4]Present address: Frontier Medicines, 151 Oyster Point Boulevard, South San Francisco, CA 94080, USA. ✉e-mail: mellman.ira@gene.com; compsagrar.laetitia@gene.com; weiruwang1999@gmail.com

production. HPK1 interferes with this pathway by phosphorylating SLP76 at residue S376, leading to the dissociation of the signaling complex followed by its ubiquitination and subsequent proteasome-dependent degradation[9]. HPK1 regulates B-cell receptor activation in an analogous mechanism through phosphorylation of B-cell linker (BLNK)[9,10].

HPK1 protein is a member of the MAP4-kinase family, with each member sharing two conserved structural features: a serine/threonine kinase domain (KD) and a citron homology domain (CHD), the two domains separated by a segment containing proline-rich regions (PRs). All MAP4K family members (MAP4K1/HPK1, MAP4K2/GCK, MAP4K3/GLK, MAP4K4/HGK, MAP4K5/KHS, MAP4K6/MINK1) contain a CHD ranging from 340 to 354 amino acids long (MAP4K1 [485–821], MAP4K2 [482–793], MAP4K3 [556–867], MAP4K4 [926–1233], MAP4K5 [506–819]). Unique to HPK1 is a caspase-3 cleavage site between PR1 and PR2 (D382–D385), marked by a DDVD motif. Caspase processing at this site leads to an N-terminal kinase domain with elevated enzymatic activity toward the JNK/MAPK pathway[11] and a C-terminal fragment that has been reported to regulate the NF-kB pathway and AICD[12]. Activation of the kinase domain is triggered by PKD- or PKA-dependent phosphorylation at residue S171, followed by auto-phosphorylation at residue T165[10,13,14]. The proline-rich region of HPK1 contains a tyrosine residue (Y381) that acts as the docking site for the kinase's substrate, SLP76, which binds to HPK1 following its phosphorylation[10].

Genetic deletion of HPK1 from T cells results in enhanced and sustained TCR signaling that leads to increased effector function[4]. Interestingly, these effects are largely replicated with a single point mutation in the kinase domain (K46E), an alteration that prevents SLP76 phosphorylation and enhances TCR signaling, as measured by pERK and cytokine production. In HPK1 K46E mice, this single mutation was sufficient to confer enhanced anti-tumor T-cell activity alone or in combination with an anti-PDL1 checkpoint inhibitor. Furthermore, pharmacologic targeting of HPK1 with a small-molecule inhibitor or use of proteolysis targeting chimera (PROTAC)-mediated degradation of HPK1 led to an improvement in the efficacy of CAR-T-cell-based immunotherapies in preclinical mouse models of hematologic and solid tumors[7].

The crystal structure of the HPK1 kinase domain has been recently described[15] in active and inactive forms, which both adopt a typical protein kinase fold. The HPK1 kinase domain is highly dynamic with significant flexibility associated with the activation segment, and a strong tendency to form domain-swapped face-to-face dimers. These structural dynamics may be essential to the regulation of the kinase function. In contrast, there is very limited information concerning the structure and function of the CHD. The CHD has been shown to function as a scaffolding protein and bind to adaptor proteins to promote JNK signaling and actin cytoskeleton rearrangement involved in lymphocyte adhesion[16]. Homodimerization of the CHD can activate MAP3K1[17] and CHD domains have been shown to directly bind kinases and promote their activity[18].

In this work, we set out to determine the crystal structure of the CHD and to understand its function related to full-length HPK1 protein. We characterize the KD-CHD interaction using biophysical methods. We then explore the functional contribution of individual domains in the context of TCR signaling upon T-cell stimulation. We demonstrate that the CHD in HPK1 acts both to enhance kinase activity and stabilize the entire protein against proteolytic degradation. Our results reveal an essential role for CHD in modulating the activity of the HPK1 kinase domain.

## Results

### Generation of HPK1 full-length and CHD proteins

We first generated recombinant proteins representing full-length HPK1 (M1-E821) (Fig. 1A) as well as just the CHD (R481-E821) in insect cells. Although both proteins eluted from the sizing column as

predominantly a single peak, the CHD construct was partially cleaved, as indicated by SDS-PAGE. N-terminal sequencing identified a protease-specific cleavage site between residues L591 and A592 (sequence in Supplementary Fig. 1A, right). Altering the residues around the cleavage site (L591A and R593A) did not succeed in preventing proteolysis. Interestingly, the full-length HPK1 protein expressed in the same host cell line was resistant to cleavage at this site, suggesting that this site is protected in the full-length structure, likely by participating in interdomain interactions.

To obtain intact CHD protein, we generated an HPK1 full-length construct that contained a thrombin cleavage site after residue V384. The full-length sample was subject to thrombin digestion during purification. Although thrombin was found not to act at the engineered site, the construct was cleaved after R427, yielding a uniform fragment containing the two PR domains at the amino terminus and an intact CHD (residues C428–E821). This construct was used for further biophysical studies.

### HPK1 CHD has a seven-bladed β-propellor fold

Although we previously reported the crystal structure of the HPK1 KD[15], a better understanding of the HPK1 function in the context of the full-length protein required an understanding of the structure of the CHD. Attempts to crystallize the full-length protein were unsuccessful, likely due to the inherent flexibility of the molecule.

Sequence analysis failed to reveal any significant homology of the HPK1 CHD to proteins with known structures. We therefore decided to pursue a single-wavelength anomalous dispersion (SAD) phasing method using selenomethionine (SeMet) labeled protein. The SeMet CHD (R481-E821) protein was expressed in a minimal medium supplemented with SeMet, then purified and crystallized as described in the "Methods" section. A complete single-wavelength anomalous diffraction dataset was collected. Eight Se sites were determined using the Autosol program (Supplementary Fig. 1B). The structure model was built and refined at 2.5 Å resolution. The overall structure of CHD exhibited a seven-bladed β-propeller fold (Fig. 1B) with each blade composed of four antiparallel β-strands (named β1, β2, β3, and β4 from the center to outside) (Fig. 1C). The seventh blade included three strands (β1, β2, and β3) on the C-terminus and one strand (β4) from N-terminus of the construct, consequently tying the polypeptide chain into a circularly folded architecture. A small domain (I567-V579) was inserted between the β3 and β4 strands of the B2 blade. The insertion partially formed a short helix but was otherwise disordered. The disordered loop harbored the proteolytic cleavage site (L591 and A592) we previously identified during overexpression in the insect cell (previous section). Interestingly, we observed that $β4$ in the B4 blade exhibited an unusual loop conformation instead of a $β$-strand. This part of the structure is at the crystal packing interface, nonetheless, the loop conformation suggests inherent flexibility. A three-dimensional surface rendering of the CHD incorporating these features is shown in Fig. 1C.

### Structural and biophysical chemical analysis reveals interdomain binding in full-length HPK1

Having determined the structure of the CHD, we next investigated its function in the context of the full-length HPK1 protein in an effort to determine whether the CHD could modulate HPK1 kinase activity. Combining the previously determined KD structure[15] with the flexibility of the intermittent region as well as the location of the SLP76 docking site in this area (Supplementary Fig. 1C), we hypothesized that the full-length HPK1 protein might fold on itself, allowing for interaction between the two domains.

To investigate the possibility of domain-domain interaction, we used hydrogen-deuterium exchange mass spectrometry (HDX-MS), a highly sensitive method for evaluating solution state dynamics with the potential to achieve single residue resolution. By comparing

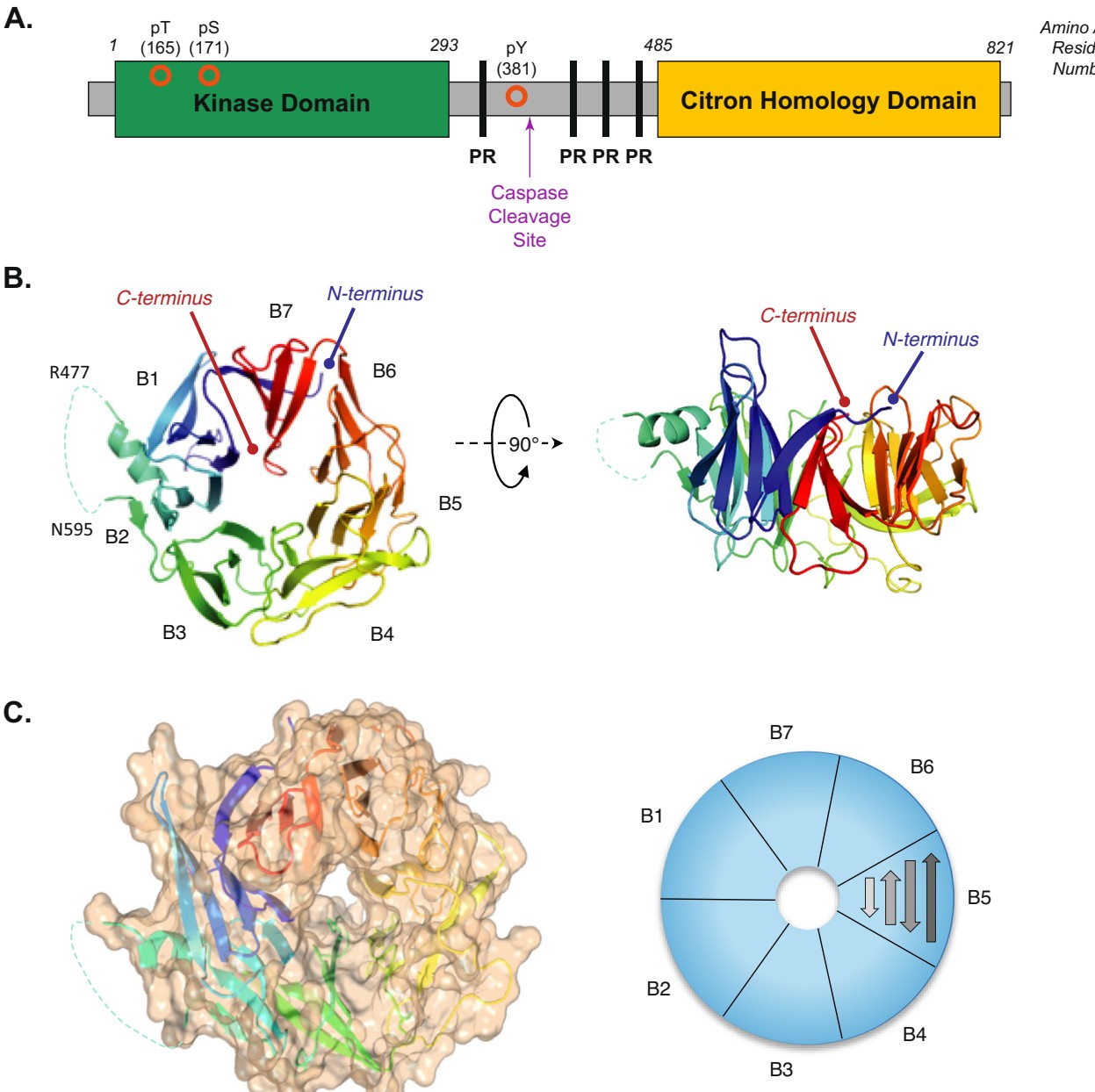

**Fig. 1 | The crystal structure of HPK1 CHD. A** Schematic of the HPK1 protein containing the kinase (green) and citron homology (yellow) domains with proline-rich regions (PR, black) and phosphorylation sites (orange circles) is indicated. **B** A ribbons diagram of HPK1 CHD structure, revealing a seven-bladed β-propeller fold. The figure displays from two viewing angles, the top view and the side view. The blades are labeled as B1–B7 in the right-hand side figure. The dotted line indicates a disordered region between B2 and B3. The ribbons are colored based on sequence order, from N-terminus (blue) to C-terminus (red). **C** Left: A molecular surface rendering of the CHD structure, with a view as shown in the left panel of (**B**). Right: A schematic depiction of the organization of the blades where each blade is composed of four antiparallel β-strands.

individual domains alone to full-length HPK1, we identified areas in both KD and CHD that were protected in the full-length protein, potentially resulting from interaction between the two domains (Fig. 2A, Supplementary Fig. 2). Summary statistics, and protection factors along with errors measured for peptides used in the final analysis are available along with uptake traces and respective error bars for all peptides in the KD and CHD (Supplementary Data 1, 2, and 3). Within the KD, the largest protection factors when compared to the full-length protein were observed in peptides covering residues 210-230, suggesting the location of the potential interaction interface. Residues D217, L221, and R222 from within this most protected region of KD along with nearby residues L170, and I173 were selected for mutation. Similar reasoning was applied to residue K558 within the

CHD. Our prior observation of protection of the β1-β2 loop in the full-length protein (Fig. 2A) indicated its involvement in the interaction with KD. We thus hypothesized that removal of this loop may affect the interaction between KD and CHD. To test this hypothesis, we generated a deletion mutant with the β1-β2 loop removed.

To assess the functional impact of KD-CHD interaction on HPK1 signaling, we generated pMSCV constructs containing HPK1 full-length sequences with polarity-modifying point mutations at each of the residues identified in the HDX-MS, or deletion of K574-594 in the case of the β1-β2 loop of the CHD. To obtain cells expressing these mutant forms of HPK1, we generated HPK1-deficient (HPK1−/−) Jurkat cells via CRISPR and used a retroviral expression system to stably express the various point mutants (Supplementary Fig. 3A–D, Supplementary

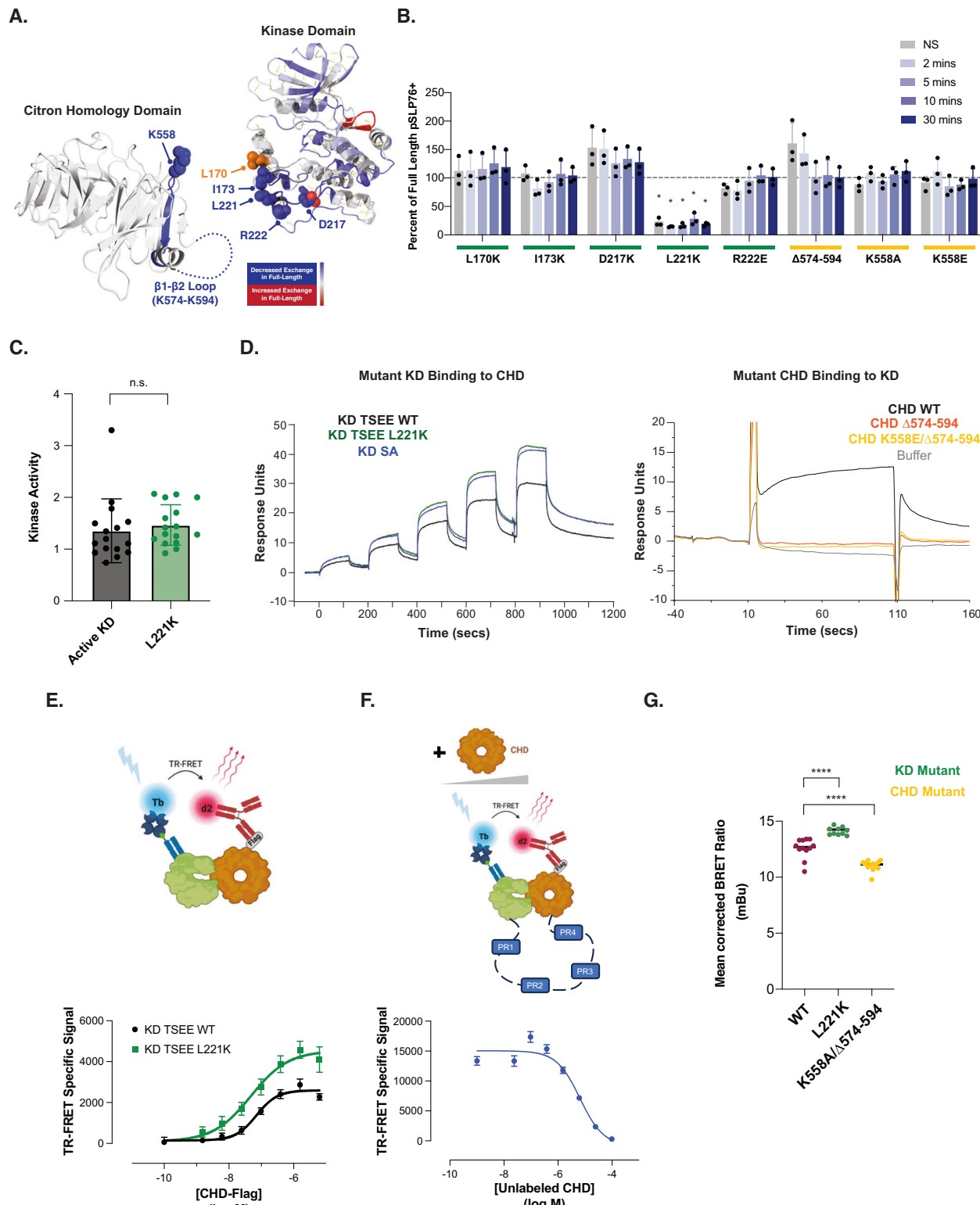

Fig. 4A). Protein expression of each of the mutant forms of HPK1 was comparable across all constructs, with only D217K demonstrating slightly reduced protein levels (Supplementary Fig. 5A, B). We then determined the ability of each to mediate the phosphorylation of the definitive HPK1 substrate SLP76 following T-cell stimulation using anti-CD3/CD28. The L221K mutation (in the KD domain) led to the most significant reduction of phosphorylated SLP76 (pSLP76), relative to full-length HPK1 across all timepoints (Fig. 2B, Supplementary Fig. 4B). This reduction of pSLP76 was not due to loss of kinase activity

as a consequence of the mutation (Fig. 2C). The D217K mutation in the KD, despite lower protein levels, trended towards increased pSLP76 at baseline, although pSLP76 levels appeared to decrease somewhat shortly after stimulation. Conversely, deletion of the β1-β2 loop (ΔK574 − K594) in the CHD had the opposite effect, demonstrating a modest trend of enhancement in pSLP76, particularly at earlier timepoints and even without stimulation (Fig. 2B). Of note, not all mutations in the presumed KD-CHD interface led to modulation of SLP76 phosphorylation, presumably because these

**Fig. 2 | Domain-domain interaction of the KD and CHD inhibits the enzymatic activity of HPK1. A** Hydrogen-deuterium exchange data for the CHD (left) and KD (right) overlaid onto their respective crystal structures. Blue indicates decreased hydrogen-deuterium exchange in the full-length form relative to the individual domain. Residues predicted to be involved in domain-domain interaction are indicated and labeled. **B** Percent of SLP76 phosphorylation after anti-CD3/CD28 stimulation in Jurkat cells expressing HPK1 mutated at the indicated residues, as measured by flow cytometry (NS indicates not stimulated). The amount of SLP76 phosphorylation is displayed relative to the amount in cells expressing full-length HPK1 at each individual timepoint. Data shown is a representative experiment of three independent experiments ($n = 3$) with means ± S.D. displayed and each construct, stimulation, and timepoint combination performed with 3 replicates. Statistical significance relative to full-length wild-type at that timepoint was determined by a two-way ANOVA with Dunnett's multiple comparisons test and is indicated above the bars. $*p = 0.0479$, $0.0428$, $0.0463$, $0.0418$, and $0.0458$ at NS, 2, 5, 10 and 30 minutes, respectively. **C** In vitro kinase assay to compare enzyme activity of KD TSEE WT or KD TSEE L221K to phosphorylate SLP76. Data shown is a representative experiment of two independent experiments ($n = 2$) with 16 technical replicates with means ± S.D. displayed. Statistical significance was assessed by a two-tailed unpaired $t$-test. n.s. not significant. **D** Binding response levels of wild-type CHD to captured KD TSEE WT, KD TSEE L221K, or KD SA (left) or mutant forms of CHD to immobilized KD TSEE WT (right) as measured by SPR. KD TSEE WT, KD TSEE L221K, or KD SA was captured by a biotinylated anti-KD antibody Fab on neutravidin immobilized sensor chips, and wild-type CHD was injected as analyte (left); KD TSEE WT was immobilized on sensor chips, and CHD mutants were injected over the surface as analyte (right). Representative sensorgrams of 4 individual sets of data are shown. **E** Measure of KD TSEE WT and KD TSEE L221K interaction with CHD using TR-FRET. Saturation binding experiments using a constant amount of recombinant KD TSEE WT or KD TSEE L221K with increasing concentrations of Flag-CHD labeled with biotinylated anti-KD Fab coupled to streptavidin-Lumi4Tb and anti-Flag Ab coupled to d2 fluorophore, respectively. The specific TR-FRET signal was calculated as follows: (total TR-FRET signal 665 nm/donor emission 620 nm) − (background TR-FRET signal 665 nm/donor emission 620 nm determined from incubating labeled KD only with anti-KD-Lumi4Tb + anti-Flag-d2). Data are mean ± S.D. of three individual experiments ($n = 3$) each performed in duplicates or triplicates. The cartoon was generated using BioRender. **F** Measure of KD-CHD interaction in the context of the full-length protein determined by competing KD-CHD interaction with increasing concentrations of recombinant untagged CHD. KD and CHD domains were labeled with TR-FRET compatible fluorophores and the specific TR-FRET was calculated as described above. Data are the mean ± S.D. of three individual experiments ($n = 3$) each performed in triplicates. The cartoon was generated using BioRender. **G** Measure of interaction between NanoLuc-KD and CHD-HaloTag in the full-length protein using NanoBRET in transfected HPK1−/− Jurkat cells. Data are means ± S.D. of two independent experiments ($n = 2$), each performed with 6 replicates. Statistical significance relative to full-length wild-type at that timepoint is indicated above the bars, using a two-way ANOVA with Dunnett's multiple comparisons test. ns not significant, $****p < 0.0001$. Illustrations in (**E**) and (**F**) were created using BioRender. For (**B**–**G**), source data are provided as a source data file.

mutations may also affect protein structure, activity and interaction with SLP76.

To determine how these modifications impacted the interaction between the KD and CHD domains, we employed surface plasmon resonance (SPR) to determine the binding of isolated domains. We previously identified significant structural changes in the KD that were associated with activation status with the T165E/S171E mutant (KD TSEE WT) mimicking the phosphorylated state of the protein and S171A (KD SA) mimicking the inactive state[15]. For these SPR assays, we used the TSEE backbone to generate the L221K KD mutant (KD TSEE L221K). We characterized the binding of WT-CHD to KD TSEE WT and KD TSEE L221K using a format where KD TSEE WT and KD TSEE L221K were indirectly captured on sensor chips. As shown in Fig. 2D, the KD TSEE L221K had significantly higher binding responses to WT-CHD compared to KD TSEE WT that may be indicative of a tighter interaction or a conformational change. As the KD TSEE L221K mutant induced a significant reduction in SLP76 phosphorylation, we hypothesized that it may behave similarly to the inactive KD (SA). Indeed, we previously observed structural differences between the active TSEE and inactive SA KD[16] and we presumed that the CHD may have differential binding activity to both forms of the kinase. Our SPR data revealed that the inactive SA form exhibited increased binding to the CHD as compared to the active TSEE form of the kinase, suggesting that interdomain binding may be regulated by kinase activation in the context of the wild-type protein (Fig. 2D).

We then analyzed the consequences of mutating the CHD on the stability of the KD-CHD interface. We chose to directly immobilize KD TSEE WT on sensor chips to minimize the non-specific binding of CHD mutants to the reference flow cell. Compared to CHD WT, CHD ΔK574-K594 with or without K558E did not exhibit significant binding to immobilized KD TSEE WT (Fig. 2D), suggesting a critical contribution of the β1-β2 loop to KD-CHD binding dynamics.

To confirm our SPR observation, we utilized an orthogonal method relying on time-resolved Förster energy transfer (TR-FRET) technology to measure the proximity between HPK1 domains. In this experiment, KD TSEE WT or KD TSEE L221K were incubated with increasing concentrations of Flag-CHD and labeled with anti-KD biotinylated Fab and anti-Flag labeled with streptavidin-Lumi4Tb (TR-FRET donor) and d2 (TR-FRET acceptor), respectively. We observed a saturating curve for both KD TSEE WT-CHD and KD TSEE L221K-CHD combinations indicative of a specific interaction between the domains (Fig. 2E). Consistently with the SPR data, the maximum binding response observed for the L221K mutant is higher than for the WT and the apparent $K_D$ of KD TSEE WT-CHD and KD TSEE L221K-CHD in this setting were 62.82 ± 9.08 and 37.92 ± 8.31 nM, respectively, suggesting a tighter interaction for KD TSEE L221K-CHD and possibly a shorter distance between the domains. Note that we were unable to obtain consistent and reliable data for the inactive KD SA in this format possibly due to protein aggregates.

While our initial TR-FRET $K_D$ determinations using separate domains did offer accurate relative comparisons of affinity across different KD-CHD pairs, it is important to note that the reported $K_D$ values were overestimated due to the specific experimental conditions required to answer our question. To confirm the presence of this KD-CHD interaction in the context of the full-length protein and measure an accurate binding affinity, we labeled full-length HPK1 bearing a Flag-tag on the C-terminus of CHD with anti-KD biotinylated Fab-streptavidin-Lumi4Tb (TR-FRET donor) and anti-Flag d2 (TR-FRET acceptor) and competed the KD-CHD interaction with unlabeled CHD. We obtained a competition curve indicative of the KD-CHD interaction within the full-length protein that allowed us to calculate a Ki equal to 5.99 μM (Fig. 2F).

To further investigate these interactions in cells, we transfected HPK1−/− Jurkat cells with constructs expressing full-length forms of HPK1 that were either wild-type, L221K (KD mutant), K558A/ΔK574-K594 (CHD mutant) bearing nanoluciferase bioluminescence resonance energy transfer (NanoBRET) compatible tags on either the N- (KD) or C-terminus (CHD) (Fig. 2G). In line with the SPR data, L221K mutant cells had increased KD-CHD binding as measured by NanoBRET relative to wild-type whereas K558A/ΔK574-K594-expressing cells displayed relatively reduced interdomain binding. Combining these observations with the pSLP76 data, we hypothesized that the strength of CHD-KD binding inversely correlated with SLP76 phosphorylation. Specifically, L221K increased the binding and/or induced a conformational change of the KD relative to the CHD, which resulted in decreased pSLP76, while ΔK574-K594 exhibited decreased binding to KD with increased SLP76 phosphorylation, indicating that KD-CHD interaction negatively modulates kinase enzymatic function.

Taken together, these results indicate the existence of KD-CHD binding dynamics in the full-length protein that has important impacts on the downstream function of the kinase domain.

### CHD is critical for optimal expression and functionality of the HPK1 kinase

To further investigate the role of the CHD in the context of full-length HPK1, we tested the impact of complete deletion of the CHD by transducing HPK1−/− Jurkat cells with retroviral constructs designed to express WT, mutant, or fragmented forms of HPK1. We observed that full-length forms of HPK1, either WT or a kinase-dead mutant (K46E), as well as CHD, both with and without the N-terminal proline-rich domains, displayed high expression levels in cells (Fig. 3A). In contrast, cells expressing forms of HPK1 without the CHD (KD only, KD + 1PR, or ΔCHD) exhibited significantly reduced protein expression, with ΔCHD having the lowest expression levels. This reduced protein expression was not attributable to differences in transfection efficiency (Supplementary Fig. 6A), retroviral production (Supplementary Fig. 6B), or transcriptional regulation (Supplementary Fig. 6C).

We then assessed how these different HPK1 forms functioned in terms of enzyme activity after TCR stimulation. As expected, the kinase-dead mutant (K46E) demonstrated a complete absence of pSLP76 compared to wild-type full-length HPK1 (Fig. 3B, C). HPK1 lacking CHD (ΔCHD, or KD + 4PRs) was highly deficient in phosphorylating SLP76 after TCR stimulation, mirroring its low protein levels and possibly an effect secondary to its protein degradation. However, while ΔCHD demonstrated the most profound reduction in protein, it was still the only deletion that produced any level of phosphorylation. The KD alone or KD + 1PR (the latter of which included the docking site Y381) showed no phosphorylation in SLP76 (Supplementary Fig. 7A). As expected, CHD fragments lacking the KD did not demonstrate SLP76 phosphorylation.

The presence of a CHD domain in all six MAP4K kinases, suggests a conserved functional role for CHD throughout the family. One such function might be to confer stability to their respective KD's either through direct interaction or protein folding. We therefore generated Jurkat cells expressing either the full-length or CHD-deleted versions of all six MAP4K family members and assessed protein expression (Fig. 3D). Notably, ΔCHD forms of each MAP4K family member (MAP4K2/GCK, MAP4K3/GLK, MAP4K4/HGK, MAP4K5/KHS, MAP4K6/MINK) exhibited dramatically reduced protein levels compared to their full-length forms, indicating that the role of supporting protein expression or stability is a shared feature for CHDs across this family of proteins.

Given this common function and the conservation of the CHD across all MAP4K members, we performed a homology assessment revealing that the CHDs of MAP4K3 and MAP4K3 exhibited the greatest percent homology (45% in CHD) to the HPK1 (MAP4K1) CHD, and MAP4K4 as having the least (22% in CHD) (Fig. 4A). Similarly, the amino acid residues in CHD that are involved in KD-CHD interactions, as revealed by above studies, show varying levels of conservation across the MAP4K family. For instance, K558 is conserved, whereas the insertion loop between the β3 and β4 strands of the B2 blade exhibits a consensus sequence of RLL(A/P)RK (residues 589-594) among MAP4K1, 2, 3, and 5. This sequence, however, is not conserved in MAP4K4, as illustrated in Fig. 4A. To determine if CHD identity was critical to HPK1 functionality or if the presence of any large domain would suffice to maintain HPK1 protein, we generated constructs to express forms of HPK1 in HPK1−/− Jurkat cells where the CHD was replaced with that of either MAP4K3 (ΔCHD + M4K3, highest homology) or MAP4K4 (ΔCHD + M4K4, lowest homology) (Fig. 4B). We observed that the protein expression of ΔCHD was rescued when the MAP4K3-CHD sequence was added, but not the MAP4K4-CHD sequence. This rescue had functional consequences for HPK1 kinase

activity as well, with ΔCHD + M4K3 demonstrating robust pSLP76 over a 60 min timecourse after TCR stimulation, closely mirroring the full-length wild-type (Fig. 4C, D). ΔCHD + MAP4K4, on the other hand, behaved similarly to ΔCHD whereby low pSLP76 correlated with low expression. These results indicate that it is not only the presence of a generic CHD that is important for the expression and function of HPK1 kinase but that the CHD identity is critical for specific kinase activity as well.

### CHD mediates docking of SLP76 to HPK1

Given the observed reduction in substrate phosphorylation by HPK1 in the absence of CHD, it seemed possible that the CHD may be required for optimal docking of SLP76. To address this, a Nanoluciferase structural complementation reporter system (NanoBit) was utilized to measure the strength and kinetics of the interaction of SLP76 with different forms of HPK1. HPK1−/− Jurkat cells were transiently transfected to express SLP76 and HPK1 with either SmBit or LgBit tags, respectively, to allow dynamic tracking of the interaction between the two over time in the presence or in the absence of TCR stimulation (Fig. 5A).

As shown in Fig. 5B, full-length wild-type HPK1 demonstrated a robust interaction with SLP76 at baseline that increased with TCR stimulation, with optimal association between 5 and 12 min. Importantly, the kinetics of the interaction correlated with SLP76 phosphorylation (Fig. 3B). In contrast, the level of interaction between ΔCHD and SLP76 was significantly less at baseline, starting at approximately 1/3 that of the full-length wild-type form despite equivalent protein expression compared to the full-length (Supplementary Fig. 8A). Notably, the extent to which TCR stimulation increased interaction with SLP76 relative to the unstimulated was comparable between full-length and ΔCHD. Note that the presence of an LgBit tag that was optimized for improved expression and stability unexpectedly conferred stability to the ΔCHD Nanobit construct. As expected, the interaction of SLP76 with HPK1 mutated for its docking residue (Y381A) was undetectable. Neither the KD nor CHD alone interacted with SLP76 (Supplementary Fig. 8B, upper panel). Interestingly, in the absence of the kinase domain (CHD+4PRs), SLP76 docking still occurred but at levels comparable to ΔCHD and was still reduced relative to the full-length (Supplementary Fig. 8B, lower panel). The inclusion of the Y381 residue was still critical despite the absence of the KD as CHD + 3PRs sequence (does not contain PR1 and Y381) did not interact with SLP76. Altogether, these data suggest that the presence of both the KD and CHD as well as the Y381 docking site are important for efficient SLP76 docking and optimal phosphorylation of SLP76 by the HPK1 KD.

To investigate if the role of CHD in SLP76 docking was direct or indirect, we conducted similar interaction studies using the ΔCHD + M4K3-CHD or ΔCHD + M4K4-CHD formats of HPK1 to compare against ΔCHD. Intriguingly, ΔCHD + M4K3-CHD behaved similarly to ΔCHD alone, with an impaired SLP76 docking compared to full-length wild-type that increased to the same extent upon TCR stimulation (Fig. 5C), regardless of the ability of MAP4K3-CHD to rescue both protein expression and HPK1 kinase function (Fig. 4C). As anticipated, no SLP76 docking was observed for ΔCHD + M4K4-CHD, correlating with minimally detectable expression of ΔCHD + M4K4-CHD in this system (Supplementary Fig. 8C). Taken together, these results suggest that the HPK1 CHD is specifically critical for optimal SLP76 docking to the HPK1 enzyme and downstream phosphorylation events.

## Discussion

Although CHDs have long since been identified as a recurring motif across a variety of proteins and species, little is known about their structure or function. For HPK1, our structural analysis has revealed the existence of a β-propeller fold, which is common to families of

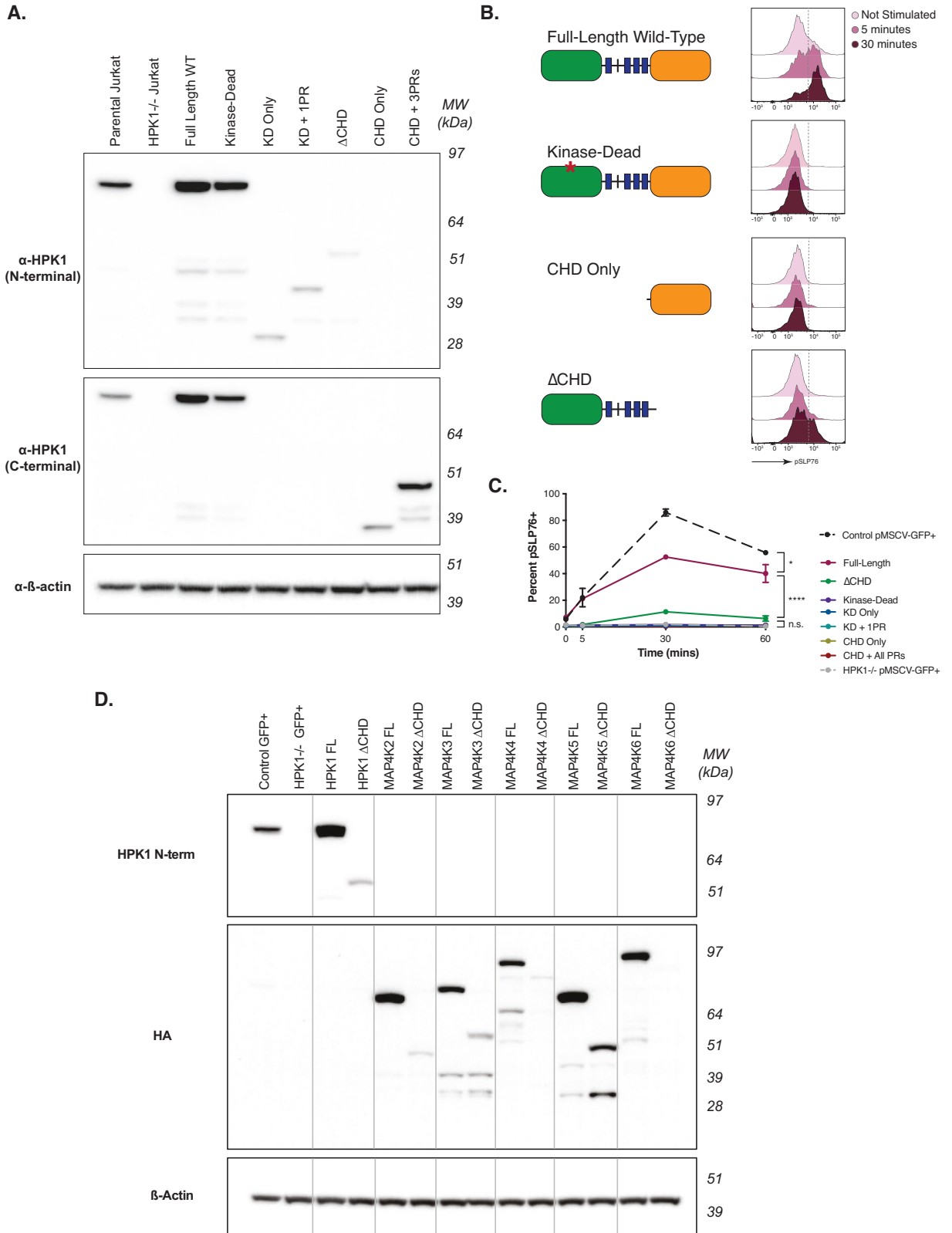

**Fig. 3 | Absence of the CHD results in deficient HPK1 kinase activity. A** Western blot showing protein levels of Jurkat cells expressing the indicated form of HPK1. **B** Phosphorylation of SLP76 over a 30-min timecourse after TCR stimulation in Jurkat cells expressing the indicated version of HPK1. **C** Percent of SLP76 phosphorylation over a one-hour timecourse in Jurkat cells expressing various forms of HPK1. Data shown in (**A**–**C**) are for one representative experiment of three independent experiments ($n = 3$) with technical triplicates and means ± S.D. displayed in

(**C**). Statistical significance was determined by two-way ANOVA with Tukey's multiple comparisons test. $p = 0.0485$ for comparison between Control GFP+ and FL WT, $p < 0.0001$ for FL WT vs ΔCHD. n.s.= not significant. **D** Protein levels of either the full-length or ΔCHD form of all members of the MAP4K family of kinases as indicated. Data in (**D**) is from a single transfection in HPK1−/− Jurkat cells. Source data are provided as a source data file.

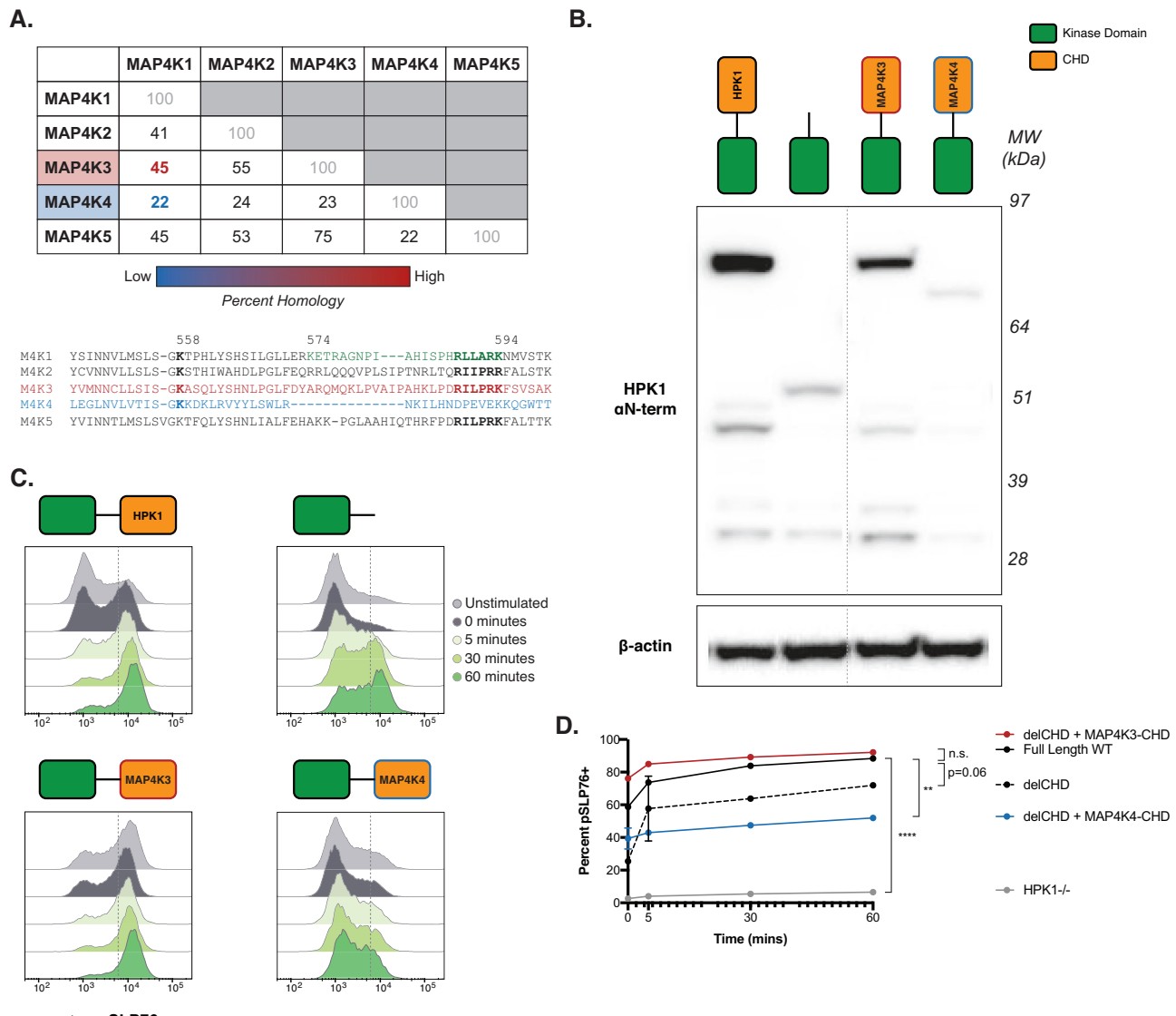

**Fig. 4 | CHD promotes HPK1 kinase activity by conferring protein stability.**
**A** Table depicting the percent homology between paired MAP4K family members specifically within the CHD of each respective protein. Red indicates the highest percent homology for HPK1 (MAP4K1), with blue indicating the lowest. The lower section shows the sequence alignment for the segment containing blade 2 of the CHD, including the insertion loop. The residue numbers indicate their corresponding positions in the HPK1 sequence. **B** Protein expression, as measured by western blot, for Jurkat cells expressing full-length HPK1, ΔCHD, ΔCHD with the CHD of MAP4K3 added to its C-terminus (43% homology), and ΔCHD with the MAP4K4-CHD (7% homology). Source data are provided as a source data file.

**C** SLP76 phosphorylation, shown as representative histograms as detected by flow cytometry, in Jurkat cells expressing the HPK1 proteins (**B**) after CD3/CD28 stimulation over a 60-min timecourse. 0 min indicates cells that were incubated with antibodies but left on ice throughout. **D** Percent of SLP76 for each cell line in (**B**) and (**C**) shown over the full timecourse. Statistical significance was determined by one-way ANOVA with Dunnett's multiple comparisons test. Data shown in (**B–D**) are for one representative experiment of three independent experiments ($n = 3$) with technical triplicates and means ± S.D. displayed in (**D**). n.s. not significant, **$p = 0.0071$, ****$p < 0.0001$. Source data are provided as a source data file.

proteins that usually display distinct repeating patterns in amino acid sequences (e.g., WD40 repeats). The number of blades can vary anywhere between 4 and 10. β-propeller proteins can display a range of function and include enzymes such as hydrolases, lyases, and oxidoreductases, as well as non-enzymatic functions including protein-protein interaction or scaffolding[19].

Notably, the HPK1 CHD sequence failed to display significant homology to previously reported β-propeller proteins therefore was not predicted to have β-propeller fold using multiple informatics tools available before we began our crystallography effort. However, a more recently developed machine learning-based algorithm, AlphaFold[20], correctly predicted the β-propeller fold for HPK1 CHD. The lack of sequence-based predictability for the CHD structures was common to

all MAP4K family members. The AlphaFold algorithm now predicted that all MAP4K CHDs adopt the β-propeller fold. Bartual et al. have recently published a crystal structure of the CHD from Rom2 protein in *Aspergillus fumigatus*[21]. CHD is located at the C-terminus of the protein and is characterized by an undefined function. The structure reveals a seven-bladed β-propeller fold, similar to that of the HPK1 CHD. Despite the structural similarities, there is a low degree of sequence homology between the CHDs of HPK1 and Rom2, with only a 20% identity – a trait commonly observed among CHDs. Collectively, these findings represent the early instances where experimental data have substantiated the CHD fold prediction by AlphaFold.

Interestingly, the AlphaFold algorithm also predicted a KD-CHD interface that is consistent with our HDX-MS data. Although AlphaFold

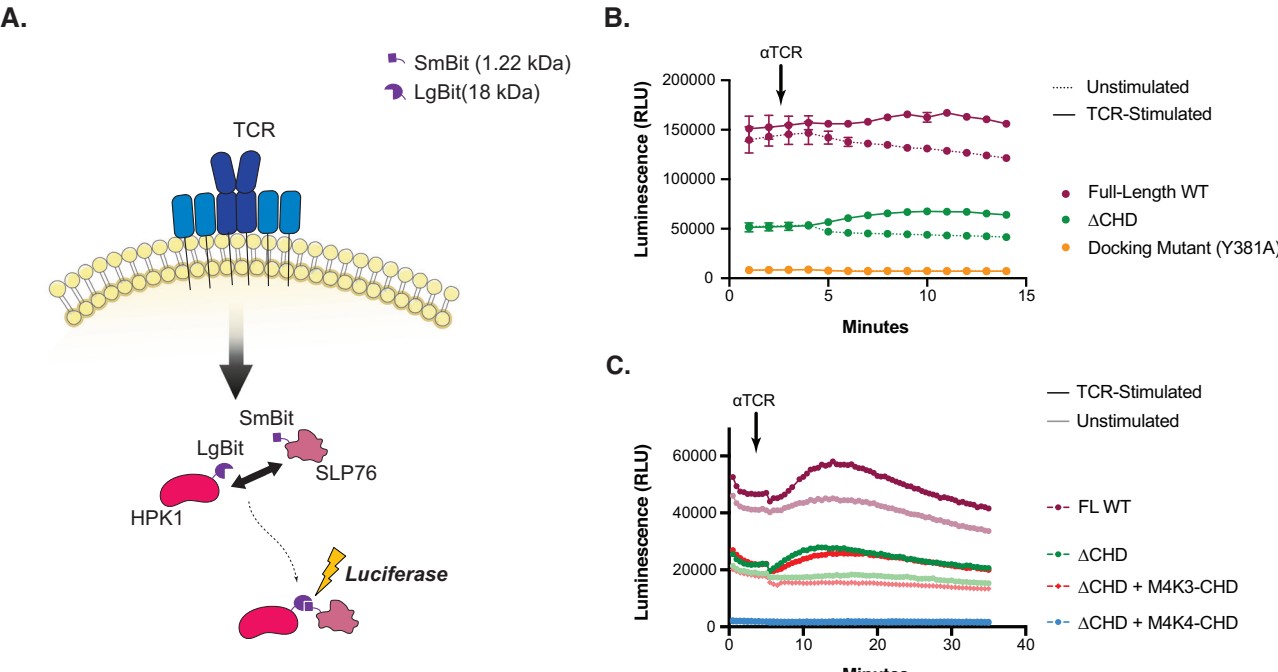

**Fig. 5 | Absence of the CHD results in reduced docking of SLP76 in live Jurkat cells. A** Schematic showing the NanoBit protein interaction assay design. HPK1 is tagged with LgBit (18 kDa) and SLP76 is tagged with SmBit (1.22 kDa). Interaction between tagged HPK1 and SLP76 in transfected Jurkat HPK1−/− results in the formation of a functional nanoluciferase protein that yields a luminescent signal. **B** Interaction between the indicated tagged HPK1 and SLP76 in real time. Luminescent signals were recorded for transfected cells stimulated with CD3/CD28 antibodies (solid lines) or isotype control (dotted lines). The arrow indicates the time of TCR stimulation. **C** Dynamics of interaction between HPK1 and SLP76, with (dark colored lines) and without (light colored lines) TCR stimulation, in forms of HPK1 where the CHD has been substituted with CHD from either MAP4K3 (red) or MAP4K4 (blue) in comparison to full-length wild-type HPK1 (burgundy) and ΔCHD (green). For (**B**, **C**), data shown is representative of three independent experiments (n = 3), each performed at least triplicates shown as means ± S.D. Source data are provided as a source data file.

claims low confidence in domain-domain interaction predictions, it helped provide justification for the empirical analysis described here. The results herein suggest that CHD facilitates the interaction of HPK1 and its substrate SLP76 in the following proposed manner: the PR domains of HPK1 enable its docking to promote subcellular localization near the TCR. Upon activation of the HPK1 kinase, dissociation of the KD-CHD releases KD for substrate interaction. Phosphorylation of the Y381 residue allows high-affinity binding of SLP76 to HPK1 mediated by CHD. CHD exerts an additional scaffolding function that likely optimizes the SLP76 phosphorylation event.

Our results suggest that CHD facilitates the interaction of HPK1 and its substrate SLP76 and thus provides insight into the spatiotemporal nature of how HPK1 acts as a negative regulator of TCR signaling. We have identified three main components of HPK1 kinase regulation by its CHD that we highlight here. The Y381 docking site has previously been shown to be critical for SLP76 phosphorylation by HPK1 and we have confirmed the importance of this residue. However, in this work, we have demonstrated an important connection between KD and CHD, and their interplay for efficient modulation of signaling. We show that both the KD and CHD contribute to efficient SLP76 binding (see Fig. 5B and Supplementary Fig. 8B). Furthermore, CHD contributes to the protein stability of the HPK1 kinase. It is still unclear how CHD contributes to the stability of the HPK1 kinase, as assessment of the proteasome degradation, lysosome degradation, and unfolded protein response did not suggest that the presence of CHD played a notable role in these pathways (Supplementary Fig. 9A–C). We cannot rule out a possible role of the interdomain PR sequences in protein stability. Of note, the PR domains contain peptide sequences that are rich in proline, glutamic acid, serine, and threonine, known as PEST sequences. It is therefore possible that either the absence of the CHD or the dissociation of the KD-CHD interaction

makes the PEST sequences open and accessible, rendering HPK1 susceptible to degradation. Lastly, CHD modulates kinase activity via interdomain interactions. Taking all these observations into account, we propose the following mechanism of action: TCR signaling initiates activation and phosphorylation of HPK1 by Zap70, which triggers the dissociation of the HPK1 kinase and citron homology domains, leaving the protein poised for SLP76 binding. Phosphorylation of Y381 further facilitates high-affinity binding of SLP76, enabling phosphorylation at its S376 residue. This key phosphorylation is further enhanced by the CHD itself, which appears to stabilize the SLP76-HPK1 interaction. These events lead to the recruitment of 14-3-3 to SLP76, 14-3-3 being a potent negative regulator of the Ras-MAPK pathway essential for TCR signaling. In addition to SLP76, HPK1 has been shown to phosphorylate GADS[22] and the members of the SLP76 family which includes SLP76, its B-cell homolog BLNK (also designated as SLP-65 or BASH), and cytokine-dependent hematopoietic cell linker (CLNK or MIST). All three proteins present a similar structure and bind to the phosphorylated residue Y381 on HPK1[10,23,24], therefore we can speculate that the role of CHD on SLP76 phosphorylation may apply to the other members of the SLP76 family in different contexts. The influence of CHD on GADS phosphorylation remains to be elucidated.

We previously reported that active and inactive KD from HPK1 forms domain-swapped dimers in solution with a KD ~ 60 μM[15]. When comparing the structure of KD-KD dimers with our HDX-MS data, we observed overlapped residues at the KD-KD and KD-CHD interfaces, suggesting that KD and CHD may compete for interaction. Our TR-FRET data using recombinant domains suggested that KD-CHD interaction had a higher affinity than KD-KD, suggesting that KD-CHD would be the preferred interaction over KD-KD. To assess whether full-length HPK1 can form dimers, we conducted analytical

ultracentrifugation (AUC) and size exclusion chromatography (SEC) experiments. At the highest solubility concentrations for AUC and comparable protein concentrations for SEC, conditions under which KD dimerization was previously evident, no dimerization of the full-length protein was observed, as confirmed by the singular peak in SEC results shown in Supplementary Fig. 1A. In addition, using our NanoBit system, we failed to detect the presence of full-length HPK1 dimers in Jurkat cells expressing tagged HPK1 (Supplementary Fig. 10A). Combined, these observations suggest that the CHD of the full-length HPK1 may interfere with the dimerization of the kinase domain by competing for interaction with key residues at the KD interface, thereby hindering dimerization in the full-length context.

A key and unique feature of HPK1 compared to other MAP4K family members is the presence of a caspase cleavage site between PR1 and PR2. What remains to be seen is how caspase cleavage impacts the dynamic of KD-KD versus KD-CHD interactions that we have identified. Chen et al. showed that caspase-3 cleavage of HPK1 altered its biochemical properties and enhanced kinase activity[25]. In this work, we similarly found kinase activation upon separation of the CHD and the KD, albeit not secondary to proteolytic cleavage. It would be interesting to further study the structural changes and dynamics that lead to kinase activation when HPK1 is cleaved by caspase-3; indeed, proteolysis may be a consequence of the conformational shift we describe here. Resolving this issue will require a detailed understanding of the KD structure in the full-length context. We believe that it is out of the scope of this paper and would warrant an additional study to address this point.

Prior literature has reported other potential roles for HPK1 CHD. For example, CHD has been suggested by early work to play a role in sensitizing T cells towards activation-induced cell death (AICD) by inhibiting NF-кB pathway[12,26,27]. In the present study, we were able to probe the dynamics of CHD regulation with TCR signaling programs intact in a relevant cellular system. HPK1 has been suggested to have additional functions in other cells, such as regulating B-cell receptor signaling in an analogous way to T cells[10,23,28]. It will be important to investigate CHD in these cell types as well to understand whether the functions we have identified are shared across immune cell types. While our work focused on the impact of CHD on TCR signaling, HPK1 is also known to be activated by prostaglandin E2 (PGE2)[14]. Thus, exploring the function of CHD in the regulation of PGE2 signaling will also be worthwhile.

Since interference with negative regulators of T-cell function, such as the PD-1 coinhibitory receptor, has proved efficacious for cancer immunotherapy, there is considerable interest in extending the approach to homeostatic inhibitors of TCR signaling such as HPK1[29]. Drug discovery efforts so far have been focused on ATP-competitive kinase inhibitors, which have encountered challenges to achieving adequate selectivity so as to avoid off-target toxicities[30–33]. Our results support additional possible strategies for HPK1 drug discovery. For example, interruption of protein-protein interactions (PPIs) with small molecules has become a feasible approach as demonstrated by the Bcl-2 inhibitor venetoclax[34]. The understanding of structure and protein-protein-interactions (PPI) that CHD involved in the cellular context opens the possibility for the discovery of PPI inhibitors to target HPK1. Additionally, targeted protein degradation is becoming a widely pursued strategy. The CHD structure reported here could enable the design of small-molecule degraders directly targeting CHD rather than the kinase ATP binding site, providing the potential for superior selectivity.

To conclude, we revealed a functional role for a CHD, by demonstrating that the CHD in HPK1 acts both to enhance kinase activity and stabilize the entire protein against proteolytic degradation. These results lend insight into a unique negative regulatory kinase and suggest avenues in the search for HPK1 inhibitors for cancer immunotherapy.

## Methods

### Expression and purification of HPK1 CHD protein for crystallization

The citron homology domain constructs (R481-E821), wild-type and R593A mutant were cloned into an intracellular BEVS expression vector with an N-terminal cleavable His6-tag and over-expressed in *sf*9 cells as previously described[15]. For expression of SeMet labeled CHD, *sf*9 cells were grown in ESF921 methionine-free medium for two days to achieve a density of $2 \times 10^6$ cells/ml before infecting with viral stock. L-selenomethionine (100 mg/L) was added after 16 h and 24 h. Cells were harvested by centrifugation about 72 h post-infection. Cell paste was resuspended in the lysis buffer (50 mM TrisCl pH 7.5, 300 mM NaCl, 10% glycerol, and 0.5 mM TCEP), homogenized, and passed through a microfluidizer three times. The lysate, clarified by centrifugation and passed through a 0.45 µm filter, was loaded onto a HisTrap FF column (GE Healthcare), and eluted with a linear imidazole gradient (20–250 mM). Fractions containing HPK1 CHD were pooled and further purified over a HiTrap SP HP cation exchange column before being treated with TEV protease to remove the His6-tag. The de-tagged protein was separated from the residual His tagged protein on a second Ni column. Finally, the CHD protein was purified on an S75 column equilibrated with 20 mM Hepes pH 7.2, 150 mM NaCl, and 0.5 mM TCEP. Pooled HPK1 CHD was concentrated at 18 mg/mL for crystallization.

### Generation of full-length HPK1, intact CHD, and KD protein for HDX-MS, SPR, and TR-FRET assay

The full-length HPK1 protein (M1-E821, T165E, S171E& R319A) with N-terminal His tag was cloned, expressed, and purified using the similar protocol as described above. The intact CHD protein was generated from the full-length construct with a thrombin site inserted after V384. Thrombin (10 U/mg) was added to the Ni elution of the HistrapFF column containing HPK1 full-length protein and the sample was dialyzed at 4 °C overnight against the following buffer: 20 mM Tris pH 7.5, 300 mM NaCl, 10% glycerol, 0.5 mM TCEP, and 2.5 mM CaCl2. The CHD-containing fragment was separated from the N-terminal His tagged kinase domain on a second Ni column and was further purified over a Hitrap SP HP column. The final buffer for the CHD protein was 20 mM Hepes pH 7.2, 150 mM NaCl, and 0.5 mM TCEP when it eluted off an S75 column. The full-length HPK1 protein (M1-E821, T165E, S171E, R319A, & Y381E) with an N-terminal His tag and a C-terminal Flag was similarly expressed and purified. Digestion of this FL protein with thrombin yielded the intact CHD with C-flag, which was further purified using Ni, Flag, and S75 columns.

Expression and purification protocol of HPK1 kinase domain (D2-N293) proteins has been described by Wu et al[15].

### Crystallization, data collection, and structure determination

Crystallization of HPK1 CHD was set up in 24-well Linbro plates using a hanging drop vapor diffusion method at 19 °C. Each well contained 500 µL of reservoir solution (0.4 M ammonium phosphate). Each drop contained 1 µL of protein solution and 1 µL of reservoir solution. Rod-shaped crystals were cryoprotected in 2 M lithium sulfate and flash-frozen in liquid nitrogen. The X-ray diffraction data were collected at Advanced Light Source (ALS) beamline 5.0.2. The inversed beam method was used to collect a single-wavelength anomalous dispersion dataset. A fluorescence scan was performed with a crystal mounted at the beam line. The X-ray wavelength was set to the peak near the Se K edge, 0.97949 Å. The diffraction images were indexed, integrated, and scaled using the program XDS[35]. AutoSol wizard[36] of PHENIX program package[37] was used for SAD phasing. AutoSol identified and refined eight out of ten Se atom positions in the asymmetric unit (Supplementary Fig. 1B). The experimental map displayed clear secondary structure features. A preliminary structure model generated by Auto-Sol was subsequently manually rebuilt in the graphics program COOT[38]

**Table 1 | Crystallography data collection and refinement statistics**

| | |
|---|---|
| PDB code | 8EEC |
| Space group | $P4_{1}2_{1}2$ |
| Unit cell | $a = b = 81.7$ Å, $c = 140.5$ Å<br>$\alpha = \beta = \gamma = 90°$ |
| Resolution range | 46.85–2.5 Å (2.64–2.50 Å) |
| Total measured reflections | 218,660 (32,537) |
| Completeness (%) | 100 (100) |
| Redundancy | 12.7 (13.3) |
| $I/\sigma$ | 18.6 (2.1) |
| Rmerge | 0.15 (1.33) |
| Resolution | 2.5 Å |
| Rwork / Rfree | 0.208/0.257 |
| Non-hydrogen atoms | 2553 |
| Water molecules | 94 |
| Average B | 57.83 |
| r.m.s.d. bond lengths | 0.005 Å |
| r.m.s.d. angles | 0.920 |
| Ramachandran (F/A/O) | 0.911/0.061/0.028 |

and refined using PHENIX.refine, in an iterative manner. The data collection and structure refinement statistics are in Table 1.

## HDX-MS experiments

HPK1 samples were prepared for labeling at different concentrations to balance different objectives of the experiment (KD: 20 µM; CHD, 40 µM; FL, 40 µM) to maximize signal intensity and avoid dimerization of the kinase domain. These samples (3 mL each) were diluted into 60 mL of deuterated labeling buffer with 10 mM Hepes, 150 mM NaCl, and a pDread of 7.1 (Kinase experiments) or 7.6 (CHD experiments) to begin the labeling experiment. The exchange process was then slowed after variable labeling times by the 1:1 addition of quench buffer, composed of 4 M GdmCl and 1 M glycine at a pH of 2.5. The material was then injected into a temperature-controlled chamber at 1 °C for online proteolytic digestion and chromatographic separation. Samples pass through a pepsin and fungal protease XIII 1:1 mixture column (NovaBioassays) before being loaded onto a BEH C8 Vanguard trap column (Waters) where they are washed for 3 min before being put online with a BEH C18 HPLC column (Waters Acquity UPLC) where they are separated by an acetonitrile gradient and then ionized by electrospray into a ThermoScientific Q-Exactive HF-x instrument for measurement of carried deuterium. Experimental manipulations were performed using a custom-built Leap HDX Pal DHR platform by Leap Technologies, Morrisville, NC.

Prior to labeling, an MS/MS experiment, used to define the retention times and identities of peptides tracked during the labeling experiments, was collected in a data-dependent top ten mode with a max fill time of 110 ms, 60k HZ resolution of parent scans, 7.5k Hz resolution on fragment spectra, and 120k Hz in ms-only mode for deuterated samples. De novo sequencing of fragment ions used Byonic® (version 3.2, Protein-Metrics Inc.) with precursor and fragment mass tolerance of 6 ppm, and then was further filtered by Byologic® (version 3.2, Protein-Metrics Inc.) using a PEP2D cutoff of >0.001 to eliminate false positive identifications. As the kinase domain was labeled under more dilute conditions than the other samples, the concentration used to prepare the peptide pool for this sample used 2x concentration. No post-translational modifications were allowed and due to relatively non-specific cleavage, there were no maximum missed cleavage sites used to identify peptides. Raw files from the

mass spectrometer were converted into an open-source format (mzXML) using the publicly available MSconvert GUI (now version 3.0.19256-a8cbe7417) and extracted ion chromatograms of deuterated peptides were produced using the ExMS program version 2[39]. Peptides were charge-state-averaged and changes in protection factors were extracted using an empirical algorithm[40,41]. Briefly, the empirical approach takes a geometric average of the ratios of time to incorporate equivalent amounts of deuterium at each condition to define a peptide-averaged protection factor for the overlapping regions of each peptide in the dataset. Each peptide used in the analysis has been included in supplementary materials, see File_S1.pdf figure for KD and File_S2.pdf figure for CHD experiments, respectively. To report a residue level protection factor, such as is shown in Fig. 2A, the smallest peptide reporting on each residue is chosen for each in the dataset. A collection summary table (File_S3.xlsx) has also been provided with relevant information about the labeling experiment, such as time-points and peptide coverage levels, the actual extracted protection factors and their respective errors, number of replicates, peptide coverage, and degree of deuterium overlap for each peptide used in the final analysis. Summary statistics for the experiment are given in Table 2.

## Generation of HPK1−/− Jurkat cells

Jurkat E6-1 cells were nucleofected with a pre-incubated complex containing recombinant Cas9 protein and CRISPR-guide RNAs (CrRNA) complex using an Amaxa 9D Nucleofector. Seven different CrRNAs were tested. Once nucleofected, cells were rested for two hours and then replated in pre-warmed media. Cells were checked for HPK1 expression by Western Blot 10 days after nucleofection (Rabbit Polyclonal, CellSignaling Technologies). Most efficient clones were then plated in as a single cell per well by limiting dilution to generate clonal populations. The efficiency of knockdown was assessed by Western Blot and clones were selected accordingly. HPK1−/− Jurkat clone 6-5 was selected for the cell-based experiments.

## Retroviral expression of HPK1 proteins in Jurkat cells

pMSCV constructs were generated in-house containing DNA to express the desired fragments and mutant forms of HPK1 along with IRES-GFP. Table 3 describes the sequences for pMSCV constructs. pMSCV DNA was transfected into HEK293T cells with Fugene 6, gag-pol, and 10A1. After 48 h, the virus-containing supernatant was harvested and filtered using a 0.45 µm filter (EMD Millipore). The virus was concentrated overnight with Retro-X Concentrator (Takara Clontech). A concentrated virus was used to infect HPK1−/− Jurkat cells in the presence of Polybrene via a spin infection. Cells were stained with Fixable Viability Dye eFluor 780 (eBioscience) and sorted to enrich for live, construct-containing cells via detection of the IRES reporter (GFP).

## Western blot

Protein expression of reconstituted cells was confirmed by Western Blot. Briefly, lysates were prepared using RIPA buffer containing cOmplete mini (Roche) and PhosSTOP (Roche). Lysates were quantified via a BCA assay (Pierce) and loaded in normalized volumes onto a 4–12% Bis-Tris SDS-PAGE gel (Novex) and run. Protein was transferred onto a nitrocellulose membrane (Bio-Rad) using a wet-transfer method. The membrane was blocked in 5% milk for 1 h and incubated with primary antibodies at 4 °C overnight (αHPK1 N-term, Cell Signaling, #4472, Dilution 1:1000; αHPK1 C-term, Santa Cruz Biotechnologies, sc-376169, Clone C-9, Dilution 1:100; αHA-Biotin, Roche, Clone 3F10, Dilution 1:500; and αB-actin HRP-Conjugated, Cell Signaling, #5125, Clone 13E5, Dilution 1:5000). Where appropriate, blots were incubated with HRP-conjugated secondary antibodies and bands detected using either an ECL or ECL Plus kit (Pierce) and visualized and imaged on an Azure 300 (Azure Biosystems).

## Table 2 | Supplementary information for HDX-MS

|  | CHD to full length | KD to full length |
|---|---|---|
| Timepoints (minutes) | 0.5, 1.37, 7.0, 38.0, 203.0, 1080.0 | |
| pH | 8 | 7.5 |
| Coverage | 87.80% | 95.50% |
| Unique Peptides | 32 | 176 |
| Redundancy | 1.67 | 8.56 |
| Average Peptide Length | 15 | 14 |
| Number of Replicates | 2 | 3 |
| Recovery (average) | 77% as estimated from controls | |

## Table 3 | pMSCV construct sequence components

| Construct name | Sequence included | Modifications |
|---|---|---|
| HPK1 full-length wild-type | M1-E821 | |
| HPK1 L170K | M1-E821 | L170K |
| HPK1 I173K | M1-E821 | I173K |
| HPK1 D217K | M1-E821 | D217K |
| HPK1 L221K | M1-E821 | L221K |
| HPK1 D217K | M1-E821 | D217K |
| HPK1 R222E | M1-E821 | R222E |
| HPK1 ΔK574-K594 | M1-E821 | ΔK574-K594 |
| HPK1 K558A | M1-E821 | K558A |
| HPK1 K558E | M1-E821 | K558E |
| HPK1 TSEE (Active) | M1-E821 | T165E/S171E |
| HPK1 SA (Inactive) | M1-E821 | S171A |
| HPK1 kinase-dead | M1-E821 | K46E |
| HPK1 KD only | M1-N293 | |
| HPK1 KD + 1PR | M1-D385 | |
| HPK1 ΔCHD | M1-G483 | |
| HPK1 CHD Only | A485-E821 | |
| HPK1 CHD + 3PRs | D382-E821 | |
| MAP4K2 full-length | M1-Y820 | |
| MAP4K2 ΔCHD | M1-K467 | |
| MAP4K3 full-length | M1-Y894 | |
| MAP4K3 ΔCHD | M1-K541 | |
| MAP4K4 full-length | M1-W1239 | |
| MAP4K4 ΔCHD | M1-T896 | |
| MAP4K5 full-length | M1-Y846 | |
| MAP4K5 ΔCHD | M1-K491 | |
| MAP4K6 full-length | M1-W1332 | |
| MAP4K6 ΔCHD | M1-V989 | |
| HPK1 ΔCHD + MAP4K3-CHD | HPK1 M1-G483 + MAP4K3 A546-Y894 | |
| HPK1 ΔCHD + MAP4K4-CHD | HPk1 M1-G483 + MAP4K4 N926-W1239 | |
| HPK1 Y381A | M1-E821 | Y381A |

### Assessment of SLP76 phosphorylation by flow cytometry

Construct-expressing cells were incubated in serum-free media overnight prior to use in stimulation assays (media contained Penicillin, Streptomycin, Glutamax, MEM Non-essential Amino Acids, HEPES, Sodium Pyruvate, and B-mercaptoethanol). Cells were pre-incubated with 10 µg/mL αCD3 (eBioscience, #14-0037-82, Clone OKT3, lot 2067619), 10 µg/mL αIgG for cross-linking (AffiniPure goat anti-mouse IgG, Jackson ImmunoResearch, #115-005-062, Lots 146203/146955), and 5 µg/mL anti-CD28 (Invitrogen, #16-0289-85, Lot 2121948, Clone

CD28.2) on ice. Cultures were then incubated in a 37 °C water bath for the indicated times and immediately fixed with Lyse/Fix (BD Biosciences) and subsequently permeabilized using Perm III Buffer (BD Biosciences). In-house generated pSLP76 antibody was used and detected with a PE-conjugated αRab F(ab')2 fragment (Cell Signaling, #8885). Data was acquired on a BD Symphony analyzer (BD Biosciences) and data was analyzed using FlowJo X (TreeStar), with composite data plotted in Prism (Graphpad Prism 9).

### In vitro kinase assay

Enzyme reactions were performed in Proxiplate Plus 384-well microplates (PerkinElmer) at 8 µL per well. Reaction buffer used contained 50 mM HEPES (pH 7.5), 0.01% Brij35 (EMD Millipore), 10 mM $MgCl_2$, and 2 mM TCEP (Sigma-Aldrich). Purified HPK1 kinase domain protein (active mutant wild-type, L221 active mutant, or inactive mutant, in-house) enzyme was prepared at a 10 nm concentration in reaction buffer and added to the plate along with the substrate mix containing 1 mM ATP and 100 nM Biotin-SLP76 (generated in-house, full-length SLP76 M1-P533, expressed in *E. coli*, containing N-terminal His- and Avi-tags) and the reaction allowed to proceed at room temperature for 1 h along with a no kinase control. Detection of phosphorylated Biotin-SLP76 was achieved with an αphospho-S376 antibody (in-house), followed by Europium labeled secondary antibody (PerkinElmer), with fluorescence read using a LANCE Ultra TR-FRET protocol on an EnVision instrument (PerkinElmer). The assay was run with 16 replicates per condition. Acceptor-to-donor TR-FRET ratio was calculated for each replicate well and plotted in Prism (GraphPad).

### Characterization of HPK1 domains interaction by SPR

Binding interactions between KD mutants and WT-CHD were evaluated using an indirect capturing format by SPR technology on a Biacore T200 instrument (Cytiva; Marlborough, MA). Briefly, NeutrAvidin (Thermo Scientific) was coupled onto four flow cells (FCs) of an SCBC HCP sensor chip (Xantec; Germany). The capture levels were ~7000 response units (RUs), using a standard amine coupling and blocking procedure recommended by the manufacturer. An anti-KD antibody Fab (generated in-house) that has minimal impact on KD binding to CHD was biotinylated and then captured via biotin-neutravidin interactions onto all four FCs, resulting in a capture level of approximately 100 RUs. WT KD and KD mutants were injected over FC2, FC3, and FC4, and the resulting capture levels for KD and KD mutants were approximately in the range of 56-73 RUs. Various concentrations of WT-CHD were diluted in running buffer (20 mM HEPES, 300 mM NaCl, 0.1% polysorbate 20, 0.5 mM TCEP, pH 7.4) and were injected over FC2, FC3, and FC4 at a flow rate of 50 µL/min for 100–120 s. All sensorgrams were generated after in-line reference cell correction followed by buffer sample subtraction. The experiments were carried out at 25 °C.

Binding interactions between KD and CHD mutants were evaluated on a direct immobilization format on a Biacore T200 instrument (Cytiva). KD was immobilized onto one FC of a Series C1 sensor chip (Cytiva) using a standard amine coupling and blocking procedure recommended by the manufacturer. The immobilization levels were approximately 300 RUs. FC1, treated in a similar manner without any proteins immobilized before blocking, was used as in-line reference FC. WT-CHD and mutants were diluted to 2 µM in running buffer (20 mM HEPES, 300 mM NaCl, 0.1% polysorbate 20, 0.5 mM TCEP, pH 7.4) and injected over immobilized KD for 90 s at a flow rate of 5 µL/min. The sensor chip was regenerated by injecting 10 mM Glycine (pH 2.0) over all FCs at a flow rate of 5 µL/min for 30 s. Sensorgrams of the interactions between ligand (KD) and analytes (CHD mutants) were generated by subtracting signals from the reference FC. The experiments were carried out at 25 °C.

## Characterization of KD-CHD interaction using TR-FRET

Binding interactions between recombinant TSEE WT and L221K KD and recombinant CHD were evaluated using TR-FRET technology. The binding assays were performed in a 384-well white microplate (Greiner) and all the reagents were diluted in assay buffer (20 mM HEPES, 300 mM NaCl, 0.1% polysorbate 20, 0.5 mM TCEP, pH 7.4). Recombinant TSEE WT or L221K KD were used at 5 μM and incubated in the presence of an increasing concentration range of recombinant Flag-CHD (1.53 nM–6.25 μM). The KD was labeled with 4 nM anti-KD biotinylated Fab (Genentech) pre-incubated with Streptavidin-Lumi4Tb (HTRF Streptavidin-Tb, Revvity) and CHD was labeled with 200 nM anti-Flag Ab-d2 (HTRF Mab anti-Flag M2-d2, Revvity, #61FG2DLB). For the $K_D$ determination of KD:CHD interaction in the context of full-length HPK1, 5 nM of HPK1 was incubated with a titration of the recombinant untagged CHD domain (23.84–97.65 μM) and labeled with 5 nM of anti-KD biotinylated Fab-streptavidin Lumi4Tb and 50 nM of anti-Flag-d2. All the reagents were incubated together for 3 h at 20 °C or ON at 4 °C and the TR-FRET signal was measured using a CLARIOstar microplate reader (BMG Labtech) with the following settings: excitation at 337 nm (200 flashes per well), donor (Lumi4Tb) emission at 620 nm, acceptor (d2) at 665 nm, integration delay 60 μs, integration time 400 μs. The specific TR-FRET signal was calculated as follows: (total FRET signal 665 nm/donor emission 620 nm) – (background FRET signal 665 nm/donor emission 620 nm determined from incubating labeled KD with anti-KD-Lumi4Tb + anti-Flag-d2). The data were plotted using GraphPad Prism version 10 and the $K_D$ was determined using a One site – Fit Ki.

## Transfection of Jurkat cells for NanoBRET or NanoBit

Jurkat cells were electroporated using SE Cell Line 4D-Nucleofector™ X Kit L (Lonza) and 4D-Nucleofactor X Unit (Lonza) per manufacturer's instructions.

## NanoBRET

Jurkat cells were transfected with constructs encoding HPK1 tagged with NanoLuc and HaloTag on the N-terminus and C-terminus, respectively, and incubated at 37 °C, 5% CO₂ for 24 h. Cells were then resuspended in Opti-MEM, 4% FCS, and seeded in a white 96-well plate at 80,000 cells/well in the presence or in the absence of HaloTag NanoBRET 618 ligand (Promega). After incubation ON at 37 °C, 5% CO₂, the NanoBRET NanoLuc substrate (Promega) was added and the donor and acceptor signal was measured using a GloMax Discover System (Promega). The NanoBRET corrected ratio was then calculated as follows: Signal ratio (618 nm/460 nm) from wells receiving HaloTag 618 ligand - Signal ratio (618 nm/460 nm) from no ligand control well.

For TCR stimulation conditions, cells were incubated with 2 μg/ml of αCD3 (BD Pharmingen) and 8 μg/ml αIgG (Jackson ImmunoResearch) and NanoBRET kinetics signal was recorded.

## NanoBit

Jurkat cells were transfected with constructs encoding HPK1 bearing NanoBit compatible tags, namely LgBit and SmBit on N- or C-terminus, and incubated at 37 °C, 5% CO₂ for 24 h. Cells were then resuspended in Opti-MEM, 2% FCS and seeded at 400,000 cells/well in a white 96-well plate. After the addition of NanoGlo Live Cell Reagent (Promega), luminescence was recorded over a defined period of time using the GloMax Discover System (Promega).

For TCR stimulation conditions, after the luminescence baseline was established, 2 μg/ml of αCD3 (BD Pharmingen) and 8 μg/ml αIgG (Jackson ImmunoResearch) were added to the cells and luminescence kinetics was recorded.

## Statistical analysis

In vitro kinase, SPR, and NanoBRET assays were analyzed by one-way ANOVA with post-hoc Dunnett's test. All statistical analyses were completed within GraphPad version 9. Differences in SLP76 phosphorylation were analyzed by two-way ANOVA. A *p-value* of 0.05 was considered to be statistically significant.

## Reporting summary

Further information on research design is available in the Nature Portfolio Reporting Summary linked to this article.

## Data availability

The crystallographic data for HPK1 CHD have been deposited into the Protein Data Bank under accession codes 8EEC. The mass spectrometry data generated in this study, along with sequence files and processed results have been deposited in the MassIVE repository under accession code MSV000094133 [https://doi.org/10.25345/C5707X03Z]. All other data supporting the findings of this study are available within the article and its supplementary files. Any additional requests for information can be directed to the corresponding authors. Source data are provided as a Source Data file with this paper. Source data are provided with this paper.

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

## Acknowledgements
We would like to acknowledge Mihiri Weerasinghe and Chamalee Gamage. We thank Christopher Sneeringer for his training on the in vitro kinase assays conducted as part of this study. We are grateful to Sascha Rutz, Avi Ashkenazi, Jon Linehan, Don Kirkpatrick, John Moffat, Jihong Yang, and John Quinn for the helpful discussions. We thank Mehraban Khosraviani for generating the biotinylated anti-KD Fab used in the SPR experiments.

## Author contributions
A.S.C., J.G., L.C.A., P.W., X.W., B.T.W., I.M., and W.W. conceived the experiments. I.L., Y.F., R.F., and P.W. performed cloning, protein expression, purification, and crystallization. W.W. determined and analyzed the crystal structure. B.T.W. performed the HDX-MS experiments and data analysis. D.W. and X.W. performed the SPR experiments and data analysis. A.S.C. and J.G. generated HPK1−/− and retroviral expression of HPK1 in Jurkat cells. A.S.C. and X.D. performed western blots. A.S.C. performed experiments on retrovirus transfected cells including SLP76 Phosphorylation experiments by flow cytometry and western blots, as well as protein validation studies and in vitro kinase assay. A.A., J.C., and L.C.A. performed NanoBRET and NanoBit assays in Jurkat cells. A.B. and L.C.A. performed the TR-FRET experiments and data analysis. A.S.C., L.C.A., I.M., and W.W. conceptualized and wrote the original draft manuscript. A.S.C., L.C.A., I.M., X.W., B.T.W., and W.W. wrote, reviewed, and edited the manuscript. All authors wrote the part of the methods corresponding to their respective experiments and reviewed the manuscript.

## Competing interests
All authors were employees of Genentech, a Member of the Roche Group, during the period in which this research was conducted. There are no more competing interests.
