## [Peer Review File · Nature Communications]

HPK1 Citron Homology Domain Regulates Phosphorylation of SLP76 and Modulates Kinase Domain Interaction DynamicsREVIEWER COMMENTS

Reviewer #1 (Remarks to the Author):

The crystal structure of HPK1 kinase domain has been reported by Dr. Wang's group as a domain-swapped dimer (Structure, 2019). In this manuscript, Chitre et al. further determined the structure of the C-terminal Citron Homology Domain (CHD) of HPK1 as a seven-bladed β -propellor fold by crystallography. Of interest, the authors identified the interaction between the HPK1 kinase domain and CHD, and they determined the putative interface residues by hydrogen-deuterium exchange mass spectrometry. The β 1- β 2 loop (K574-K594) within CHD of HPK1 was identified to contact the kinase domain, and its mutation results in activation of HPK1 kinase activity. The authors propose that CHD provides stability to HPK1 protein in cells and contributes to the docking of its substrate SLP-76 in TCR signaling. Overall, the interaction between CHD and kinase domain of HPK1 is an interesting observation. There are several major concerns listed below.

Major concerns:

1. The crystal structure of the Citron-Homology Domain for Rom2, the guanine nucleotide exchange factor Rho, has recently been reported also to adopt a seven-bladed β -propeller fold (Bartual et al, PNAS, 118:e2110298118, September 20, 2021). The publication should be cited and both CHD structures should be compared.
2. The authors concluded that intramolecular interaction of kinase domain with CHD decreases HPK1 kinase activity and increases its protein stability. Does intermolecular interaction between KD and CHD occur, resulting in protein stabilization of KD protein? Furthermore, what is the underlying regulatory mechanism of HPK1 protein stability by CHD?
3. In Fas-mediated T cell apoptosis, caspase 3-mediated cleavage of HPK1 at D385 results in separation of CHD from the kinase domain as well as activation of HPK1 kinase activity (Chen et al., Oncogene 1999). The published results should be compared with the authors' results.
4. The authors have previously reported that HPK1 kinase domain (KD) forms dimer. Therefore, it is important to investigate whether CHD competes with or promotes the dimer formation of HPK1 KD, a possible scenario during Fas-mediated apoptosis.
5. In Figure 2B, mutations of multiple residues implicated in mediating the interaction between KD and CHD did not always yield anticipated results. The authors should discuss those negative results.
6. In Figures 4C/D and 5A, the CHD of MAP4K3 was capable of functionally substituting for the CHD of HPK1. Does this imply MAP4K3 could form heterodimer with HPK1? Some studies on this issue would be helpful.

Minor points:

1. In Introduction section, the CHD of different MAP4Ks should be specified: TNIK/MAP4K7 (line 97), MAP4K3 (line 99) and MAP4K4 (line 100).
2. The authors claimed that “N-terminal sequencing identified a protease-specific cleavage site between residues L591 and A592” (lines 117-118), but the result was not provided.
3. Please verify the statement: “A small domain (I567-V579) was inserted between β 3 and β 4 strands of the B2 blade” (lines 145-146) and label this small domain in Figure 1. Of note, “deletion of K574-594 in the case of the β 1- β 2 loop of CHD” was stated elsewhere (lines 178-179 and 190-191)
4. Figure S2 does not contain “Protection factors along with errors in their estimation are shown in SIHX1” as stated (line 171).
5. The description of MAP4K CHD deletion mutant constructs should be included.
6. In Discussion section, the authors concluded that “This key phosphorylation is further enhanced by the CHD itself, which appears to stabilize the SLP76-HPK1 interaction.” This seems to be an over interpretation of the results as the induced interaction of SLP76-HPK1 may simply due to the induction of HPK1 protein levels by CHD.

Reviewer #2 (Remarks to the Author):

This study reports on the first experimental structure of the citron homology domain (CHD) from a MAP4K protein (HPK1). Using this new structure and the earlier determined crystal structure of the HPK1 kinase domain (KD), determined by the same group in 2019 and reported in Structure, the authors propose a mechanism based on which the full-length protein binds and phosphorylates SLP76 at S376. HPK1 is an important kinase that appears to control T-cell activation, and it is one of the negative intracellular breaks that limits signaling after TCR activation. Because of this, it is a “hot” kinase and it is an important pharmaceutical target: blocking the brake may have beneficial effects in immunotherapy against cancer for example. Thus it would be important to explore the specifics of full-length (FL) HPK1 regulation.

My overall opinion is that this is an interesting study about the role of CHD in HPK1-mediated SLP76 phosphorylation. Based on the experimental data, it appears that the activity and substrate binding capacity of FL HPK1 is modulated by CHD. Conceptually, this is not that surprising since we know of several examples when ser/thr kinase activity of upper-tier MAPK cascade components are

regulated this way: upstream signaling events relieve auto-inhibition exerted by auxiliary domains distinct from the KD. Because MAP3Ks and MAP4Ks relay signals from various receptors, therefore these upper-tier kinases have extra regulatory domains that greatly vary and the regulatory mechanisms are complex and are often unique to a MAP3K or MAPK4 subgroup. The authors show another nice example of this type of multi-domain dependent, complex regulation.

The conclusion of the manuscript is that CHD regulates phosphorylation of SLP76 and modulates “kinase domain interaction dynamics”. The CHD and KD are linked by a flexible linker (~200aa) and contains Y381 which, after tyrosine-phosphorylation, is a docking site for SLP76 substrate binding. It is argued that 1) CHD is indirectly involved by SLP76 phosphorylation because there is a direct correlation between CHD-KD interaction and negative regulation of kinase activity. Moreover, it is also argued that 2) CHD provides stability to HPK1 in cells and 3) it also contributes to the docking of its substrates. These conclusions are reached based on experiments carried out using deletion or point mutation bearing HPK1 constructs studied in relevant cell-based tests (Jurkat cells), which is a great strength of this work. These experiments clearly suggest an important role for CHD. However, the concrete mechanisms underlying 1-3 are not satisfactorily explored. These three likely have distinct mechanisms, albeit CHD dependent, and further work is required to correctly explore them.

Major comments:

1) The SPR data on KD and CHD binding is not adequate (Fig. 2D and F). These data is meant to show that 1) L211K KD (active) binds better than KD (inactive), 2) different mutant CHDs bind KD worse than WT CHD, and 3) WT CHD binds inactive KD better than active KD. All these statements are based on only single concentration measurements. I do not think that it is adequate to use SPR for this. The analyte (CHD) was used always in 2 μ M concentration as stated in the M&M section. Without knowing the K_d (steady-state dissociation constant, binding affinity) of the interaction, it is impossible to design a robust SPR experiment to compare the binding affinity of different analytes this way (e.g., due to precision in protein concentration measurements or pipetting errors). Moreover, it is impossible to judge whether the binding is specific and the expected stoichiometry between the ligand and analyte can indeed be achieved as expected based on single point measurements.

There are also some inconsistencies between the sensorgrams obtained with different ligand capture methods. For the chip prepared by direct amine coupling of CHD (Fig 2.D right panel) the expected analyte response, used at the same concentration (2 μ M), normally should be higher compared to a chip surface on which the ligand is indirectly captured (left panel) - unless direct amine coupling interferes with some of the binding functionalities of the ligand. For the same WT CHD analyte concentration direct amine coupling gives only RU ~ 10, while indirect capture gives RU ~ 80. The authors should discuss the limitations of their SPR experiments, or preferably carry out a more rigorous, quantitative analysis based on multiple analyte concentration measurements.

The captured ligand (KD) on Fig. 2D (right panel) is 320-340 RU (per M&M section) and 2 μ M CHD gives close to half maximum RU (80-100), meaning that 2 μ M would be close to the K_d (binding affinity) of this domain-domain interaction in trans. This appears to be too strong for an interaction that naturally occurs in cis, and it would be important to discuss how the authors envision the

relationship between CHD and KD structurally. Does CHD contact and block KD activity or allosterically affects it through the contacts whose strength they were exploring with SPR experiments in trans?

2) Do the authors believe that intact CHD-KD contacts (e.g. those explored in HDX-MS) make the FL HDK1 more compact compared to when the “interface” residues are subjected to polarity-modifying point mutations or loop deletions? If this is the case then I believe that this needs to be directly addressed (e.g., SAXS or size-exclusion chromatography) because SLP76 phosphorylation data is not informative in this regard because that is also effected by the “docking” function of CHD. The role of CHD regarding to pY381 binding is touched upon but there is no discussion on how this is related to the contact area that is explored by HDX-MS. Would there be a different surface on CHD which would be involved in docking? Another different issue that needs further exploration? How does CHD stabilize protein level concretely?

3) CHD and KD interdomain cooperation regarding protein stability and SLP76 phosphorylation seems to be specific since not all MAP4K CHD could replace HPK1 CHD. Do the authors see some specific sequence conservation pattern for residues that they believe mediate CHD-KD contacts that could be have specific roles? I believe that this needs to be at least discussed beyond global sequence conservation comparisons.

4) The interpretation of the HDX-MS data needs to be more carefully handled. The authors published a paper on the KD of HPK1. This was a domain-swapped face-to-face dimer, involving those regions that the authors in this manuscript indicate to be protected in FL HPK1 compared to the KD construct. The authors argued in their 2019 paper (Structure) that the huge dimer interface (~170-230) is likely physiologically relevant. This new study implicates residues within this region to be important contacts for the CHD. In the M&M section the authors mention that they carried out the exchange reaction at 1 mM KD, 2 mM CHD and 2 mM FL samples (because the original 20 or 40 mM stocks were diluted 20-fold into deuterated labeling buffer). Based on the Kd of dimerization given in their 2019 paper (~60 μ M) the KD sample was likely a dimer under the exchange conditions. I believe that it is not known what the oligomeric state of FL HPK1 is under the used exchange conditions. Therefore, I think that the authors need to discuss the pitfalls of this HDX-MS experiments. Knowing of this dimerization, the interpretation of the protection factors is not that straightforward, because differences between KD and FL HPK1 may simple arise due to their different dimerization tendencies, too.

Minor comments:

1) The manuscript only addresses HPK1 mediated SLP76 phosphorylation on one specific site. Are there more known bona fide substrates of HPK1? Does CHD play similar roles in other substrates, too. This would be important to be discussed in order to increase the scope of the authors' findings, at least in the discussion.

- 2) I do not see the relevance of the caspase cleavage site on Figure 1A, but indication of domain or important construct boundaries by amino acid numbering would be useful.
- 3) It is not clear how to read color coding on Fig. 2B. NS (not significant) – gray. Different time points are colored in blue but no 0 min. Which data is statistically significant?
- 4) The plots on Fig. 3B are not discernible. Are all these constructs needed? What is the relevance of KDomain + DDVD
- 5) What's the relevance of delCHD+LgBit on Fig. 4D? Do all data points have error bars on this panel?
- 6) Fig. S2 shows the protection factors of HDX-MS experiment. This figure would be helpful if it would contain residue level information, showing the full sequence next to this data with secondary structural elements indicated. How would "increased exchange in full-length" (red residues on Fig.2A) appear on this plot? It would also be useful to show the same protection data for CHD, since Fig. 2A shows protection of some of the residues for this domain, too.
- 7) For SPR data panels please clarify what the ligand was in the experiment and how it was captured on the surface in the figure legends, too, to avoid confusion. (Based on how it is presented it took some time to figure out that CHD was the analyte in all experiments).

Reviewer #3 (Remarks to the Author):

Given that HPK1 is a negative regulator of T-cell signalling, its inhibition could be an important component of anti-cancer immunotherapies. Chitre et al. have made an excellent contribution to understanding the regulation of HPK1, and it could have an important role in developing inhibitors of the enzyme that function outside the ATP-binding pocket of the kinase domain. The authors have married structural methods (crystallography and HDX-MS) with innovative biophysical measurements of interactions and signal quantitation in cells to arrive at a plausible model of the mechanism by which HPK1 gives rise to phosphorylated SLP76. Following on from their earlier work that elucidated the structure of the HPK1 kinase domain, in this manuscript, the authors have determined the structure of the C-terminal CHD of HPK1, and they find that both the kinase and CHD domains have structures that closely match the alphafold predictions for the structures of these domains. This manuscript represents the first experimental confirmation of the alphafold prediction that the C-terminal domain of HPK1 has a WD40 fold. One aspect of the manuscript that is confusing concerns its connection with their earlier report of the structure of the HPK1 kinase domain (KD). In the earlier structure manuscript, the authors had a crystal structure that showed a domain swapped dimer for the kinase domain, and the authors showed that a fraction of the purified protein is

present in solution as dimers. It might have been inferred from the previous work that the dimers in the crystal are relevant to the function of HPK1 in cells. The current manuscript does not mention the proposed dimeric arrangement, nor does it remark on how this might relate to the alphaFold prediction of the full-length protein. If the authors now regard the domain-swapped dimer to be a crystal artefact, it would be helpful if they could comment on this. If they still believe that this dimeric arrangement is important for the function of HPK1, it would be helpful if they could speculate on this dimer in the context of the structure of the two domains that make up HPK1 and the relationship of them to each other as inferred by their HDX-MS.

Most of the manuscript concerns the role of the CHD domain in the cellular function of HPK1. One innovation that the authors have brought to the effort to understand this regulation is that they have used an immortalized T cell line, to investigate the role of CHD-mediated regulation in cells with intact TCR signalling programs. Among the most innovative aspects of the work is that the authors have been able to infer the interface between the KD and CHD, based on HDX-MS and the results of HDX-MS-guided mutagenesis. The authors show that the CHD is necessary for stability of the enzyme in cells, and they show that the CHD domain contributes to optimal docking of the substrate SLP76 to HPK1.

There are several minor points that require some attention:

The legend for Fig. 1c seems to be garbled:

“(C) A molecular surface rendering of the CHD structure view from opposite from relative to the right figure in (A)”

The panel in A is a domain organisation bar diagram. Not a structure. As such, it is not clear what the orientation referred to in A might mean.

A better legend might be:

“(C) Left: A molecular surface rendering of the CHD structure, with a view as shown in the left panel of (B). Right: A schematic depiction of the blades organization. Each blade is composed of four antiparallel beta-strands

Lines 80-82: “The proline-rich region of HPK1 contains a tyrosine residue (Y381) that acts as the docking site for the kinase’s substrate, SLP76, which binds to HPK1 following its phosphorylation.”

This statement is based on a publication from about 20 years ago showing that pTyr381 acts as the binding site for the SH2 domain of SLP76. However, the authors cite no reference for their statement. I think the correct reference should be Sauer et al 10.1074/jbc.M106811200. Mentioning the pTyr/SH2 interaction here would make the statement on lines 244-245 easier to understand [“KD alone or KD+1PR (the latter of which included the docking site Y381) showed no phosphorylation in SLP76 (Figure S6A”)]

Lines 146-147: “A small domain (I567-V579) was inserted between β 3 and β 4
146 strands of the B2 blade”

In the alphafold model, this helix is predicted along with a second helix, and this second helix is part of the predicted interface with the kinase domain. Is HDX-MS consistent with this?

In Figure 2, only the L221K mutation affected the downstream phosphorylation (Figure 2B). In the alphafold model, this is at the interface. Is it in HDX-MS?

Lines 166-169: The HDX-MS results are described here. Apart from the painted ribbon diagram, there is only Figure S2A that represents any HDX-MS results. Figure S2A is simply insufficient to understand the HDX-MS results. The following should be given:

1. There should be a table giving the sequences for each peptide identified
2. For each peptide, there should be a deuterium uptake for each time point
3. There should be means and standard deviations for all of the measurements
4. How many replicates were there for each time point?
5. How is the protection factor defined? There is a reference to a previous publication (reference 38), but it would be helpful to summarise here.
6. What were the criteria that were used to select peptides with significant protection?
7. Were there also de-protections observed for any peptides?
8. Have the HDX-MS data been deposited to the PRIDE database?

Probably many of these data were present in Files S1 and S2 that are referred to in the text, however, no such files have been made available to the reviewers, if they were uploaded by the authors.

It is essential to have the full peptide information, so that it would be possible to check the protection for the region that alphafold predicts to wrap around the CHD to hold the KD and CHD together (residues 292-369). The alphafold model shows confident prediction for part of this region forming a fifth strand along blade 1 of the CHD. Given how long the alphafold model has been available, it may be that the authors have already done this analysis. It would be helpful if the authors could comment on this prediction and whether there is any significant difference in the HDX-MS that would be either consistent or inconsistent with the prediction. While alphafold predictions commonly have good agreement with folded kinase and WD40 domains, interdomain interactions are more mixed, and the authors can contribute to evaluating these predictions.

Lines 172-174: “Our prior observation of protection of the β 1- β 2 loop in the full-length protein (Figure S1A) indicated its involvement in the interaction with KD.”

It is not clear what the authors mean by their “prior observation.” HDX for this loop does not seem to have been referred to previously in this manuscript. Was this a previously published observation? If so, there should be a reference. The authors probably mean the β 1- β 2 loop of the blade 2 (where residue 558 is), but they should do this explicitly. The alphafold model shows two helices in this loop. Is this consistent with the absolute HDX protection values for the region corresponding to the second helix in the loop? The alphafold mode predicts that this loop interacts with the kinase domain. How do the differences in the protection factors agree with this prediction?

Lines 290-292: The authors state: “Interestingly, in the absence of the kinase domain (CHD+4PRs), SLP76 docking still occurred but at levels comparable to Δ CHD and was still reduced relative to the full-length (Figure S7B, lower panel).”

The lower figure does not seem to show the nanobit measurement for the full-length HPK1, only CHD+3PRs or CHD+4PRs. The upper panel probably has full-length kinase dead HPK1, but it means comparing one line from the lower panel with one line from the upper panel. Even if this comparison is made, it is not clear that CHD+4PRs shows reduced luminescence relative to the full-length HPK1. This is critical to the next statement “Altogether, these data suggest that presence of both the KD and CHD as well as the Y381 docking site are important for efficient SLP76 docking and optimal phosphorylation of SLP76 by the HPK1 KD.” Obviously, the kinase domain is necessary for SLP76 phosphorylation, but the observations do not show that it is important for docking.

Lines 521-522: The kinase assay methods is confusing. It currently refers to “active mutant wild-type.” It is not clear what is “mutant wild-type.” This needs to be rephrased in a more coherent way. I am guessing that an active mutant wild-type means phosphorylation mimic mutant. This should be stated.

Line 524: There is a reference to something called “Biotin-SLP76.” However, there is nothing in the manuscript that explains what this is. Full-length SLP76 biotinylated? A SLP76 peptide? How is it biotinylated. One purpose of the methods is to facilitate repeating the work. This is not possible without explicit methods.

Response to Referees Letter

We would like to express our gratitude to the reviewers for their insightful comments and questions. Our detailed responses are included below, in blue for your convenience.

Reviewer #1 (Remarks to the Author):

The crystal structure of HPK1 kinase domain has been reported by Dr. Wang's group as a domain-swapped dimer (Structure, 2019). In this manuscript, Chitre et al. further determined the structure of the C-terminal Citron Homology Domain (CHD) of HPK1 as a seven-bladed β -propellor fold by crystallography. Of interest, the authors identified the interaction between the HPK1 kinase domain and CHD, and they determined the putative interface residues by hydrogen-deuterium exchange mass spectrometry. The β 1- β 2 loop (K574-K594) within CHD of HPK1 was identified to contact the kinase domain, and its mutation results in activation of HPK1 kinase activity. The authors propose that CHD provides stability to HPK1 protein in cells and contributes to the docking of its substrate SLP-76 in TCR signaling. Overall, the interaction between CHD and kinase domain of HPK1 is an interesting observation. There are several major concerns listed below.

Major concerns:

1. The crystal structure of the Citron-Homology Domain for Rom2, the guanine nucleotide exchange factor Rho, has recently been reported also to adopt a seven-bladed β -propeller fold (Bartual et al, PNAS, 118:e2110298118, September 20, 2021). The publication should be cited and both CHD structures should be compared.

We have addressed this by citing the mentioned publication within the discussion as well as speaking to the comparison of the two structures (lines 370-376). This section now highlights that there is low sequence homology (20% identity) between HPK1 and Rom2 Citron-Homology Domains, which is common among CHDs.

2. The authors concluded that intramolecular interaction of kinase domain with CHD decreases HPK1 kinase activity and increases its protein stability. Does intermolecular interaction between KD and CHD occur, resulting in protein stabilization of KD protein? Furthermore, what is the underlying regulatory mechanism of HPK1 protein stability by CHD?

The previous SPR and NanoBRET data as well as the newly generated SPR and TR-FRET data presented in Figure 2D-G support an intermolecular interaction between KD and CHD. As discussed in the manuscript, we observed decreased stability of the kinase domain compared to full-length protein, suggesting that CHD plays a role in stabilizing the protein.

Also in response to the Reviewer's comment, additional experiments were performed to assess whether Δ CHD was degraded via the proteasome, lysosome, or unfolded protein response (UPR) pathways. Proteasome degradation was measured by looking at protein levels after MG132 treatment, lysosome-based degradation after chloroquine treatment, and UPR by measuring Δ CHD in both soluble and insoluble cellular fractions. Δ CHD did not accumulate in any of the three experimental settings, thus the mechanism of why Δ CHD has reduced protein stability is still unclear. Data was added to **Figure S9A-C** and the following section was incorporated to the discussion:

“It is still unclear how CHD contributes to the stability of the HPK1 kinase, as assessment of the proteasome degradation, lysosome degradation, and unfolded protein response did not suggest that the presence of the CHD played a notable role in these pathways (**Figure S9A-C**). We cannot rule out a possible role of the interdomain PR sequences in protein stability. Of note, the PR domains contain peptide sequences that are rich in proline, glutamic acid, serine and threonine, known as PEST sequences. It is therefore possible that either the absence of the CHD or the dissociation of the KD-CHD interaction, make the PEST sequences open and accessible, rendering HPK1 susceptible to degradation.”

3. In Fas-mediated T cell apoptosis, caspase 3-mediated cleavage of HPK1 at D385 results in separation of CHD from the kinase domain as well as activation of HPK1 kinase activity (Chen et al., Oncogene 1999). The published results should be compared with the authors' results.

We agree with the reviewer that it would be interesting to look at the effect of caspase 3-mediated cleavage on HPK1 structural dynamics. We have added the following points into the revised Discussion section to compare our results with those previously published, although we did not directly study the effects of caspase-3.

“Chen et al. showed caspase-3 cleavage of HPK1 altered its biochemical properties and enhanced kinase activity. In this work, we similarly found kinase activation upon separation of the CHD and the KD, albeit not secondary to proteolytic cleavage. It would be interesting to further study the structural changes and dynamics that lead to kinase activation when HPK1 is cleaved by caspase-3; indeed, proteolysis may be a consequence of the conformational shift we describe here. Resolving this issue will require a detailed understanding of the KD structure in the full-length context. We believe that it is out of scope for this paper and would warrant an additional study to address this point.”

4. The authors have previously reported that HPK1 kinase domain (KD) forms dimer. Therefore, it is important to investigate whether CHD competes with or promotes the dimer formation of HPK1 KD, a possible scenario during Fas-mediated apoptosis.

We agree with the reviewer and we were also interested in addressing this point. We engineered some constructs to express either KD or delta-CHD with NanoBit compatible tags in order to study the domain homodimerization in HPK1^{-/-} Jurkat cells. We first optimized the orientation of the tags in COS7 cells and were unable to measure any signal for KD dimerization. We moved forward with the delta-CHD optimized constructs in HPK1^{-/-} Jurkat cells and were unable to show any dimerization even upon TCR stimulation. We observed a time dependent interaction between HPK1 and SLP76 as expected. Given the importance of CHD in stabilizing the KD (Figure 4), we cannot rule out that this negative result was the consequence of a lack of sufficient expression of the SmBit-delta CHD. Unlike the LgBit tag (used throughout the manuscript for delta CHD:SLP76 interaction) that was engineered to improve expression and stability, SmBit likely did not confer stability to the SmBit-delta-CHD construct. At this point, we cannot conclude whether KD homodimerization is relevant in cells because of technical challenges due to the protein instability.

5. In Figure 2B, mutations of multiple residues implicated in mediating the interaction between KD and CHD did not always yield anticipated results. The authors should discuss those negative results.

The authors agree that not all mutations at the KD-CHD interface identified by HX-MS resulted in diminished SLP76 phosphorylation, as shown in Figure 2B. This observation suggests several potential additional scenarios:

1. The mutation might not significantly impair the CHD-KD interaction.
2. The disruption/modulation of CHD-KD interaction might enhance KD activity, but simultaneously make the KD:SLP76 interaction less optimal.

In summary, these mutations can have diverse effects on protein structure, activity, and their interactions with SLP76, leading to varied outcomes on SLP76 phosphorylation. This point was added to the result section (lines 202-204)

6. In Figures 4C/D and 5A, the CHD of MAP4K3 was capable of functionally substituting for the CHD of HPK1. Does this imply MAP4K3 could form heterodimer with HPK1? Some studies on this issue would be helpful.

We thank the reviewer for raising an interesting point. In our previous study (Wu, 2011), we identified kinase domain dimerization through two distinct experiments: 1) analytical ultracentrifugation (AUC) and 2) size exclusion chromatography (SEC). In the current study, we attempted AUC using the full-length HPK1 protein. However, we did not observe dimerization, even at the highest achievable protein concentration for the full-length protein. Similarly, when performing SEC at concentrations comparable to those used for the kinase domain, only a monomer peak was detected (see updated, Figure S1A). These observations suggest that the recombinant full-length protein has a significantly diminished propensity to homo-dimerize, if at all.

We were also interested in evaluating whether HPK1 full length was capable of forming homodimers in cells. To address this, we conducted NanoBit experiments using HPK1^{-/-} Jurkat cells transfected with constructs expressing HPK1-LgBit and SmBit-HPK1 in the presence or absence of TCR stimulation. As a positive control, we monitored HPK1:SLP76 interaction. We did not observe any homodimerization of HPK1 regardless of the activation status of Jurkat cells.

We also used NanoBRET technology as an orthogonal method to assess HPK1 homodimerization. In this setting, constructs expressing HPK1 tagged with either a NanoLuc or a Halotag were expressed in HPK1^{-/-} Jurkat cells, labeled with NanoBRET Nano-Glo substrate and HaloTag NanoBRET 618 Ligand and NanoBRET signal was measured in real time in the presence or absence of TCR stimulation. Similarly to the data obtained with NanoBit, we confirmed the HPK1:SLP76 interaction upon TCR stimulation (left panel) but were unable to measure a consistent NanoBRET signal between full length HPK1 (right panel).

Together, these results do not support homodimerization of full length HPK1 in T cell lines. Therefore, it is highly unlikely that HPK1 could form heterodimers with MAP4K3.

We incorporated this data to the manuscript (Figure S1A and Figure S10) and highlighted the different behavior of the isolated kinase domain vs. the full-length domain in the discussion (lines 423-432).

Minor points:

1. In Introduction section, the CHD of different MAP4Ks should be specified: TNIK/MAP4K7 (line 97), MAP4K3 (line 99) and MAP4K4 (line 100).

In response to the reviewer's comment, the following has been added to the Introduction section:

"All MAP4K family members (MAP4K1/HPK1, MAP4K2/GCK, MAP4K3/GLK, MAP4K4/HGK, MAP4K5/KHS, MAP4K6/MINK1) contain a CHD ranging from 340-354

amino acids long, (MAP4K1 (485-821), MAP4K2 (482-793), MAP4K3 (556-867), MAP4K4 (926-1233), MAP4K5 (506-819)).”

2. The authors claimed that “N-terminal sequencing identified a protease-specific cleavage site between residues L591 and A592” (lines 117-118), but the result was not provided.

The cleavage site has now been added to **Figure S1A**.

3. Please verify the statement: “A small domain (I567-V579) was inserted between β 3 and β 4 strands of the B2 blade” (lines 145-146) and label this small domain in Figure 1. Of note, “deletion of K574-594 in the case of the β 1- β 2 loop of CHD” was stated elsewhere (lines 178-179 and 190-191)

Figure 1B was edited accordingly.

4. Figure S2 does not contain “Protection factors along with errors in their estimation are shown in SIHX1” as stated (line 171).

The authors have corrected the figure and file numbering and associated legends to accurately reflect the contents of Figure S2, File S1, File S2, and File S3.

5. The description of MAP4K CHD deletion mutant constructs should be included.

To address the reviewer’s comment, Supplementary Table 3 has been added to describe sequences and/or modifications for all pMSCV constructs utilized in experiments included in the manuscript.

6. In Discussion section, the authors concluded that “This key phosphorylation is further enhanced by the CHD itself, which appears to stabilize the SLP76-HPK1 interaction.” This seems to be an over interpretation of the results as the induced interaction of SLP76-HPK1 may simply be due to the induction of HPK1 protein levels by CHD.

The interpretation mentioned in the discussion is based on the results shown in Figure 5B. For generating this data, we took advantage of the NanoBit that relies on the use of LgBit and SmBit tags. As highlighted in the results section, the presence of the LgBit tag unexpectedly improved the expression of the tagged constructs and thus all HPK1 constructs expressed equally in the SLP76 recruitment assay (Figure S8A). Therefore, we believe that our interpretation is correct.

Reviewer #2 (Remarks to the Author):

This study reports on the first experimental structure of the citron homology domain (CHD) from a MAP4K protein (HPK1). Using this new structure and the earlier determined crystal structure of the HPK1 kinase domain (KD), determined by the same group in 2019 and reported in Structure, the authors propose a mechanism based on which the full-length protein binds and phosphorylates SLP76 at S376. HPK1 is an important kinase that appears to control T-cell activation, and it is one of the negative intracellular breaks that limits signaling after TCR activation. Because of this, it is a “hot” kinase and it is an important pharmaceutical target: blocking the brake may have beneficial effects in immunotherapy against cancer for example. Thus it would be important to explore the specifics of full-length (FL) HPK1 regulation.

My overall opinion is that this is an interesting study about the role of CHD in HPK1-mediated SLP76 phosphorylation. Based on the experimental data, it appears that the activity and substrate binding capacity of FL HPK1 is modulated by CHD. Conceptually, this is not that surprising since we know of several examples when ser/thr kinase activity of upper-tier MAPK cascade components are regulated this way: upstream signaling events relieve auto-inhibition exerted by auxiliary domains distinct from the KD. Because MAP3Ks and MAP4Ks relay signals from various receptors, therefore these upper-tier kinases have extra regulatory domains that greatly vary and the regulatory mechanisms are complex and are often unique to a MAP3K or MAP4K subgroup. The authors show another nice example of this type of multi-domain dependent, complex regulation.

The conclusion of the manuscript is that CHD regulates phosphorylation of SLP76 and modulates “kinase domain interaction dynamics”. The CHD and KD are linked by a flexible linker (~200aa) and contains Y381 which, after tyrosine-phosphorylation, is a docking site for SLP76 substrate binding. It is argued that 1) CHD is indirectly involved by SLP76 phosphorylation because there is a direct correlation between CHD-KD interaction and negative regulation of kinase activity. Moreover, it is also argued that 2) CHD provides stability to HPK1 in cells and 3) it also contributes to the docking of its substrates. These conclusions are reached based on experiments carried out using deletion or point mutation bearing HPK1 constructs studied in relevant cell-based tests (Jurkat cells), which is a great strength of this work. These experiments clearly suggest an important role for CHD. However, the concrete mechanisms underlying 1-3 are not satisfactorily explored. These three likely have distinct mechanisms, albeit CHD dependent, and further work is required to correctly explore them.

Major comments:

1) The SPR data on KD and CHD binding is not adequate (Fig. 2D and F). These data are meant to show that 1) L211K KD (active) binds better than KD (active), 2) different mutant CHDs bind KD worse than WT CHD, and 3) WT CHD binds inactive KD better than active KD. All these statements are based on only single concentration measurements. I do not think that it is adequate to use SPR for this. The analyte (CHD) was used always in 2 μ M concentration as stated in the M&M section. Without knowing the K_d (steady-state dissociation constant, binding affinity) of the interaction, it is impossible to design a robust SPR experiment to compare the binding affinity of different analytes this way (e.g., due to precision in protein concentration measurements or pipetting errors). Moreover, it is impossible to judge whether the binding is specific and the expected stoichiometry between the ligand and analyte can indeed be achieved as expected based on single point measurements.

The authors thank the reviewer for the comment. It has been very challenging to determine accurate affinity for CHD binding to KD using SPR. As shown in the updated Figure 2D, despite extensive efforts spent to optimize assay conditions, the obtained binding curves between CHD and KD (mutants) did not follow the expected simple 1:1 binding pattern, and therefore no reliable binding kinetics and affinity parameters could be derived. The deviation from the 1:1 binding model may have various causes including non-specific binding of CHD or small amounts of aggregates present in the tested CHD protein although other factors such as unknown binding interactions cannot be ruled out. Nonetheless, the observations we made earlier that 1) KD mutants showed higher binding responses to CHD than active KD, and 2) CHD mutants showed lower binding responses to KD than WT CHD hold true when we repeated the experiment with multiple concentrations, as shown in the updated Figure 2D.

To strengthen our argument, we added a series of TR-FRET experiments (Figure 2E-F) that should address the reviewer's comment.

There are also some inconsistencies between the sensorgrams obtained with different ligand capture methods. For the chip prepared by direct amine coupling of CHD (Fig 2.D right panel) the expected analyte response, used at the same concentration (2 μ M), normally should be higher compared to a chip surface on which the ligand is indirectly captured (left panel) - unless direct amine coupling interferes with some of the binding functionalities of the ligand. For the same WT CHD analyte concentration direct amine coupling gives only RU \sim 10, while indirect capture gives RU \sim 80. The authors should discuss the limitations of their SPR experiments, or preferably carry out a more rigorous, quantitative analysis based on multiple analyte concentration measurements.

The reviewer was correct that we used two SPR assay formats to characterize KD mutants and CHD mutants binding, and observed different binding response levels with the two formats. For the first format, WT KD and mutants were captured by an anti-KD antibody that is believed to have minimal impact on KD binding to CHD; for the second format, WT KD was directly immobilized on sensor chips. The first format was preferred as indirect assay format offers several advantages over the direct immobilization approach including the ligand protein is not chemically modified, ligand molecules can be oriented on the surface in a more uniform manner, and ligand binding properties can be preserved without the deleterious effects of regeneration conditions. Since the anti-KD antibody has minimal impact on KD binding to CHD, and that we observed much lower binding responses with the 2nd format, it is likely that the immobilization chemistry modifies KD binding properties. The reason that we chose the 2nd format to assess binding interactions between CHD mutants and KD was because a CHD mutant had significant non-specific binding to the reference flow cell when it was tested with the 1st format, making the comparison invalid. We agree with the reviewer to mention the 2 formats and to explain the rationale for choosing one format over the other one (refer to Figure 2 results section and Figure legend). Meanwhile, we have conducted experiments with multiple concentrations and included the data in the manuscript (refer to the response to the comment above and updated Figure 2D).

The captured ligand (KD) on Fig. 2D (right panel) is 320-340 RU (per M&M section) and 2 μ M CHD gives close to half maximum RU (80-100), meaning that 2 μ M would be close to the K_d (binding affinity) of this domain-domain interaction in trans. This appears to be too strong for an interaction that naturally occurs in cis, and it would be important to discuss how the authors envision the relationship between CHD and KD structurally. Does CHD contact and block KD activity or allosterically affects it through the contacts whose strength they were exploring with SPR experiments in trans?

As explained in the responses above, the obtained SPR data are heterogeneous and complicated by other factors including non-specific binding and possible aggregates in the proteins, which precluded determination of reliable kinetics and affinity parameters. We agree with the reviewer that SPR assay has its limitations in characterizing challenging binding interactions between KD and CHD. For this reason, we implemented an orthogonal assay using TR-FRET and confirmed the binding of KD to CHD across a wide range of concentrations (Figure 2E) as well as in the context of the full-length protein (Figure 2F) and determined an affinity of 5.99 μ M. We hypothesize that CHD allosterically affects the position and function of KD and as the reviewer correctly pointed out, we were exploring the strength of the contacts in trans using SPR and more recently in cis using TR-FRET. Collectively, our comparison of binding of CHD with KD versus L221K KD using SPR and TR-FRET (Figure 2D and 2E)

suggests that we modify the binding affinity and possibly induce a conformational change that would support the allostery hypothesis.

2) Do the authors believe that intact CHD-KD contacts (e.g. those explored in HDX-MS) make the FL HPK1 more compact compared to when the “interface” residues are subjected to polarity-modifying point mutations or loop deletions? If this is the case then I believe that this needs to be directly addressed (e.g., SAXS or size-exclusion chromatography) because SLP76 phosphorylation data is not informative in this regard because that is also effected by the “docking” function of CHD. The role of CHD regarding to pY381 binding is touched upon but there is no discussion on how this is related to the contact area that is explored by HDX-MS. Would there be a different surface on CHD which would be involved in docking? Another different issue that needs further exploration? How does CHD stabilize protein level concretely?

We are grateful for the reviewer's insightful suggestion. We concur that the molecular envelope might undergo alterations when KD and CHD are separated. So far, our SEC analyses have not revealed significant differences between the mutant and wild-type full-length proteins. However, we acknowledge that more comprehensive techniques, such as SAXS, could yield further clarity on this matter. We do not currently have access to this technology. While these investigations fall beyond the scope of our current manuscript, we could explore them in subsequent studies.

pY381 is located in the proline-rich region and thus we do not believe there is any connection with the KD:CHD interface. Our NanoBit data also suggest that there is no other surface on CHD that is involved in docking as a single point mutation (Y381A) suffices to completely abrogate the recruitment of SLP76 to HPK1 (Figure 5B).

It is still unclear how CHD stabilizes the protein level. Additional experiments were performed to assess whether Δ CHD was degraded via the proteasome, lysosome or unfolded protein response (UPR) pathways. Proteasome degradation was measured by looking at protein levels after MG132 treatment, lysosome-based degradation after chloroquine treatment, and UPR by measuring Δ CHD in both soluble and insoluble cellular fractions. Δ CHD did not accumulate in any of the three experimental settings, thus the mechanism of why Δ CHD has reduced protein stability is still unclear. Data has been added to **Figure S9A-C** and the following section was incorporated to the discussion:

“It is still unclear how CHD contributes to the stability of the HPK1 kinase, as assessment of the proteasome degradation, lysosome degradation, and unfolded protein response did not suggest that the presence of the CHD played a notable role in these pathways (**Figure S9A-C**). We cannot rule out a possible role of the interdomain PR sequences in protein stability. Of note, the PR domains contain peptide sequences that are rich in proline, glutamic acid, serine and threonine, known as PEST sequences. It is therefore possible that either the absence of the CHD or the dissociation of the KD-CHD interaction, make the PEST sequences open and accessible, rendering HPK1 susceptible to degradation.”

3) CHD and KD interdomain cooperation regarding protein stability and SLP76 phosphorylation seems to be specific since not all MAP4K CHD could replace HPK1 CHD. Do the authors see some specific sequence conservation pattern for residues that they believe mediate CHD-KD contacts that could be have specific roles? I believe that this needs to be at least discussed beyond global sequence conservation comparisons.

We thank the reviewer for the insightful suggestion. We conducted a more detailed analysis of the sequences of the CHD residues under investigation. The analysis indicated a short sequence within the insertion loop of blade 2 that is conserved among a subset of MAP4K members, specifically M4K1, 2, 3, and 5, but not M4K4 (revised **Figure 4A**). This finding aligns well with our results demonstrating enhanced stability with the M4K3-CHD chimeric construct, as opposed to the M4K4-CHD chimeric construct. Such a distinction in stability may be attributed to a differential level of homology. Additionally, our Western blot analysis of Jurkat cells expressing a loop-deletion variant of HPK1 revealed that the level of protein expression is comparable to that of the wild-type (Figure S4). This suggests that the insertion loop may not be the sole contributor affecting protein stability in cells. We postulate that the mechanism by which CHD confers stabilization to the protein complex likely involves additional elements beyond the consensus sequence within the insertion loop. It is conceivable that other regions of the protein domain play a pivotal role in this stabilization process, potentially through intricate interactions or conformational dynamics that warrant further investigation.

We added a section to address the reviewer's comment (lines 294-301).

4) The interpretation of the HDX-MS data needs to be more carefully handled. The authors published a paper on the KD of HPK1. This was a domain-swapped face-to-face dimer, involving those regions that the authors in this manuscript indicate to be protected in FL HPK1 compared to the KD construct. The authors argued in their 2019 paper (Structure) that the huge dimer interface (~170-230) is likely physiologically relevant. This new study implicates residues within this region to be important contacts for the CHD. In the M&M section the authors mention that they carried out the exchange reaction at 1 mM KD, 2 mM CHD and 2 mM FL samples (because the original 20 or 40 mM stocks were diluted 20-fold into deuterated labeling buffer). Based on the K_d of dimerization given in their 2019 paper (~60 μM) the KD sample was likely a dimer under the exchange conditions. I believe that it is not known what the oligomeric state of FL HPK1 is under the used exchange conditions. Therefore, I think that the authors need to discuss the pitfalls of this HDX-MS experiments. Knowing of this dimerization, the interpretation of the protection factors is not that straightforward, because differences between KD and FL HPK1 may simply arise due to their different dimerization tendencies, too.

In looking into the reviewer's comment, we realized that there was a typo in the original submission where micro-molar was written as milli-molar. The HDX experiments were conducted at concentrations below the dissociation constant of the homodimer. To be sure, in experiments that were not discussed, multiple concentrations of HPK1 KD were used in parallel labeling experiments to determine whether any changes were due to dimer dissociation that could confound the analysis of FL HPK1 to KD. This typo has been corrected, and the authors thank the reviewer for noticing this discrepancy.

Minor comments:

1) The manuscript only addresses HPK1 mediated SLP76 phosphorylation on one specific site. Are there more known bona fide substrates of HPK1? Does CHD play similar roles in other substrates, too. This would be important to be discussed in order to increase the scope of the authors' findings, at least in the discussion.

The reviewer raised an interesting point. We have edited the discussion to elaborate on this point (lines 411-417).

2) I do not see the relevance of the caspase cleavage site on Figure 1A, but indication of domain or important construct boundaries by amino acid numbering would be useful.

The authors feel that inclusion of the caspase cleavage site on the schematic is relevant to points included in the discussion. It may help the reader visualize that the caspase-mediated cleavage results in two different fragments containing each of the domains, KD and CHD, and thus this has been maintained. To address the reviewer's comment, amino acid number boundaries for the two domains were added.

3) It is not clear how to read color coding on Fig. 2B. NS (not significant) – gray. Different time points are colored in blue but no 0 min. Which data is statistically significant?

We would like to clarify that “NS” denotes the “not stimulated” condition, instead of “not significant” as suggested by the reviewer. We have clarified this in the figure legend of **Figure 2**.

4) The plots on Fig. 3B are not discernible. Are all these constructs needed? What is the relevance of KDomain + DDVD?

The authors would like to thank the reviewer for highlighting this as this was a mislabeling. The “KDomain + DDVD” corresponds to “KD + 1PR” and has been included to convey how presence of one PR (with the docking residue Y381) versus all PRs impacts kinase expression/function. The labeling has been unified across **Figures 3A, 3B, and 3C**.

5) What's the relevance of delCHD+LgBit on Fig. 4D? Do all data points have error bars on this panel?

The authors thank the reviewer for highlighting this as this LgBit construct was used for assay optimization and was not intended to be kept in the final figure. The LgBit construct has been removed from **Figure 4D**.

All constructs and timepoints were completed in triplicate and all data points were included to generate this plot. The graph error bars are shorter than the size of the symbol and thus are not visible.

6) Fig. S2 shows the protection factors of HDX-MS experiment. This figure would be helpful if it would contain residue level information, showing the full sequence next to this data with secondary structural elements indicated. How would “increased exchange in full-length” (red residues on Fig.2A) appear on this plot? It would also be useful to show the same protection data for CHD, since Fig. 2A shows protection of some of the residues for this domain, too.

Individual peptide plots have been added to the manuscript and so that any interested reader can manually inspect each peptide if desired. The coloring on Figure 2A is normalized such that the largest reduction in exchange is colored dark blue and the largest increase in exchange is colored dark red. As a consequence, only a very small amount of destabilization is required to produce the deep red because the magnitudes of PF changes in the direction of increased exchange were very small. Directly, these red regions appear as negative deflections from the log(PF) in Figure S2.

7) For SPR data panels please clarify what the ligand was in the experiment and how it was

captured on the surface in the figure legends, too, to avoid confusion. (Based on how it is presented it took some time to figure out that CHD was the analyte in all experiments).

We agree with the reviewer and have included the information in the figure legend of **Figure 2D**.

Reviewer #3 (Remarks to the Author):

Given that HPK1 is a negative regulator of T-cell signalling, its inhibition could be an important component of anti-cancer immunotherapies. Chitre et al. have made an excellent contribution to understanding the regulation of HPK1, and it could have an important role in developing inhibitors of the enzyme that function outside the ATP-binding pocket of the kinase domain. The authors have married structural methods (crystallography and HDX-MS) with innovative biophysical measurements of interactions and signal quantitation in cells to arrive at a plausible model of the mechanism by which HPK1 gives rise to phosphorylated SLP76. Following on from their earlier work that elucidated the structure of the HPK1 kinase domain, in this manuscript, the authors have determined the structure of the C-terminal CHD of HPK1, and they find that both the kinase and CHD domains have structures that closely match the alphafold predictions for the structures of these domains. This manuscript represents the first experimental confirmation of the alphafold prediction that the C-terminal domain of HPK1 has a WD40 fold. One aspect of the manuscript that is confusing concerns its connection with their earlier report of the structure of the HPK1 kinase domain (KD). In the earlier structure manuscript, the authors had a crystal structure that showed a domain swapped dimer for the kinase domain, and the authors showed that a fraction of the purified protein is present in solution as dimers. It might have been inferred from the previous work that the dimers in the crystal are relevant to the function of HPK1 in cells. The current manuscript does not mention the proposed dimeric arrangement, nor does it remark on how this might relate to the alphafold prediction of the full-length protein. If the authors now regard the domain-swapped dimer to be a crystal artefact, it would be helpful if they could comment on this. If they still believe that this dimeric arrangement is important for the function of HPK1, it would be helpful if they could speculate on this dimer in the context of the structure of the two domains that make up HPK1 and the relationship of them to each other as inferred by their HDX-MS.

We would like to thank the reviewer for bringing up a very valid point. We generated additional data to address this point and added the following section in the discussion to clarify our current hypothesis concerning HPK1 dimerization in the context of the full-length protein:

“We previously reported that active and inactive KD from HPK1 forms domain-swapped dimers in solution with a $K_D \sim 60\mu\text{M}$ ¹⁵. When comparing the structure of KD-KD dimers with our newly generated HX data, we observed overlapped residues at the KD-KD and KD-CHD interfaces, suggesting that KD and CHD may compete for interaction. Our TR-FRET data using recombinant domains suggested that KD-CHD interaction had a higher affinity than KD-KD, suggesting that KD-CHD would be the preferred interaction over KD-KD. To assess whether full-length HPK1 can form dimers, we conducted analytical ultracentrifugation (AUC) and size exclusion chromatography (SEC) experiments. At the highest solubility concentrations for AUC and comparable protein concentrations for SEC, conditions under which KD dimerization was previously evident, no dimerization of the full-length protein was observed, as confirmed by the singular peak in SEC results shown in **Figure S1A**. In addition, using our NanoBit system, we failed to detect the presence of full length HPK1 dimers in Jurkat cells expressing tagged HPK1 (**Figure S10A**). Combined, these observations suggest that the CHD of the full-length HPK1

may interfere with the dimerization of the kinase domain by competing for interaction with key residues at the KD interface, thereby hindering dimerization in the full-length context.”

While we do not have direct evidence confirming the functional relevance of the HPK1 kinase domain dimer in cellular context, we cannot rule out the possibility that the dimeric form may be relevant under specific conditions, such as in Fas-mediated T cell apoptosis (as discussed in lines 433-440). Our findings suggest that the full-length HPK1 and the isolated kinase domain represent distinct functional states. A more thorough exploration of the interplay and respective roles of these states presents an attractive direction for future studies.

Most of the manuscript concerns the role of the CHD domain in the cellular function of HPK1. One innovation that the authors have brought to the effort to understand this regulation is that they have used an immortalized T cell line, to investigate the role of CHD-mediated regulation in cells with intact TCR signalling programs. Among the most innovative aspects of the work is that the authors have been able to infer the interface between the KD and CHD, based on HDX-MS and the results of HDX-MS-guided mutagenesis. The authors show that the CHD is necessary for stability of the enzyme in cells, and they show that the CHD domain contributes to optimal docking of the substrate SLP76 to HPK1.

There are several minor points that require some attention:

The legend for Fig. 1c seems to be garbled:

“(C) A molecular surface rendering of the CHD structure view from opposite from relative to the right figure in (A)”

The panel in A is a domain organisation bar diagram. Not a structure. As such, it is not clear what the orientation referred to in A might mean.

A better legend might be:

“(C) Left: A molecular surface rendering of the CHD structure, with a view as shown in the left panel of (B). Right: A schematic depiction of the blades organization. Each blade is composed of four antiparallel beta-strands

We have corrected the figure legend for **Figure 1C** to align with the reviewer’s suggestion.

Lines 80-82: “The proline-rich region of HPK1 contains a tyrosine residue (Y381) that acts as the docking site for the kinase’s substrate, SLP76, which binds to HPK1 following its phosphorylation.”

This statement is based on a publication from about 20 years ago showing that pTyr381 acts as the binding site for the SH2 domain of SLP76. However, the authors cite no reference for their statement. I think the correct reference should be Sauer et al 10.1074/jbc.M106811200.

Mentioning the pTyr/SH2 interaction here would make the statement on lines 244-245 easier to understand [“KD alone or KD+1PR (the latter of which included the docking site Y381) showed no phosphorylation in SLP76 (Figure S6A”)]

We thank the reviewer for pointing this out. We added the reference to the relevant statement.

Lines 146-147: “A small domain (I567-V597) was inserted between β 3 and β 4 146 strands of the B2 blade”

In the alphafold model, this helix is predicted along with a second helix, and this second helix is part of the predicted interface with the kinase domain. Is HDX-MS consistent with this?

It was not our intention to confirm or negate the AlphaFold model. Significant protection was observed in the region comprising residues 567-597, consistent with the proposed helix by AlphaFold. The region of increased protection was large, spanning from roughly 555-613. Directly, we can conclude that something occurs in this region of the protein leading to a significant reduction of exchange rates through new protection factors in the full-length molecule explained either with or without concomitant change in secondary structure. See **File S3** for relevant peptides.

In Figure 2, only the L221K mutation affected the downstream phosphorylation (Figure 2B). In the alphaFold model, this is at the interface. Is it in HDX-MS?

According to the HX-MS results, L221 was part of the most protected region when comparing full-length HPK1 and CHD/KD only. Thus, we chose to make the L221K mutation as we expected this residue to be at the KD/CHD interface.

This is mentioned in the following results section:

“To assess the functional impact of KD-CHD interaction on HPK1 signaling, we generated pMSCV constructs containing HPK1 full-length sequences with polarity-modifying point mutations at each of the residues identified in the HX-MS, or deletion of K574-594 in the case of the β 1- β 2 loop of the CHD.”

Lines 166-169: The HDX-MS results are described here. Apart from the painted ribbon diagram, there is only Figure S2A that represents any HDX-MS results. Figure S2A is simply insufficient to understand the HDX-MS results. The following should be given:

1. There should be a table giving the sequences for each peptide identified

Please see answer to Question 3 below.

2. For each peptide, there should be a deuterium uptake for each time point

Please see answer to Question 3 below.

3. There should be means and standard deviations for all of the measurements.

In two collections of plots, each as **Files S1 and S2** that were inadvertently left out of the submission, all of this information is given in a visual format. For KD vs FL, $n=3$, stddevs are given by error bars (though many are so small they blend with the actual curve) and for CHD vs FL, $n=2$ - here error bars in the SI figures represent the range of measurements. Replicate counts are given in the SI HDX table.

4. How many replicates were there for each time point?

Please see answer to Question 3 above.

5. How is the protection factor defined? There is a reference to a previous publication (reference 38), but it would be helpful to summarise here.

Briefly, an empirical method is used whereby the geometric mean of all ratios of time to achieve equivalent deuterium incorporation in either condition is taken over the range of deuterium incorporation values that overlap between two conditions being tested. As there are error bars,

for any uptake trace, three curves may be generated, one that traces the mean of each measurement, a second that represents the high valued error bar and a third representing the low valued error bar, as described in a separate manuscript. As each condition now has three traces that describe the uptake, we can take the mean empirical PF as the observed PF and the two extremes to represent an error for the mean value. Significant values were those whose mean PF exceeded the standard error estimated by this method. These methods are both cited and a brief discussion has been added to the manuscript Materials & Methods section as well.

6. What were the criteria that were used to select peptides with significant protection?

Please see the response to Question 5 above.

7. Were there also de-protections observed for any peptides?

There were a small number of peptides deprotected in the dataset. 95-101(FCGAGSL), representative of residues 98,99 (GS) have an empirical $\log(\text{PF})$ of -0.17, confirmed by 97-103, and 97-105 (**Files S1-S3**).

8. Have the HDX-MS data been deposited to the PRIDE database?

All raw MS data is available upon request and uptake traces are available as supplementary figures and files associated with this manuscript. We did not deposit the data to the PRIDE database as it would require significant efforts due to file sizes and our corporate intranet policies.

Probably many of these data were present in Files S1 and S2 that are referred to in the text, however, no such files have been made available to the reviewers, if they were uploaded by the authors.

The authors have corrected this and included all relevant **Files S1-S3**.

It is essential to have the full peptide information, so that it would be possible to check the protection for the region that alphaFold predicts to wrap around the CHD to hold the KD and CHD together (residues 292-369). The alphaFold model shows confident prediction for part of this region forming a fifth strand along blade 1 of the CHD. Given how long the alphaFold model has been available, it may be that the authors have already done this analysis. It would be helpful if the authors could comment on this prediction and whether there is any significant difference in the HDX-MS that would be either consistent or inconsistent with the prediction. While alphaFold predictions commonly have good agreement with folded kinase and WD40 domains, interdomain interactions are more mixed, and the authors can contribute to evaluating these predictions.

These residues (292-369) did not exist in either the KD or CHD constructs used in this work and our concern in the manuscript was to determine where HX rates changed comparing KD and CHD alone to full length. Therefore, this information is not included in the manuscript. A peptide that has not been included in this work (291-336) did show a small amount of protection in roughly 20% of the exchange sites. The remaining sites exchanged before the first timepoint, suggestive of low protection. This being said, it is difficult to predict exchange rates from structure alone, and unclear whether this information corroborates the model or simply suggests rapidly reorganizing structure with modest protection.

Lines 172-174: “Our prior observation of protection of the β 1- β 2 loop in the full-length protein (Figure S1A) indicated its involvement in the interaction with KD.”

It is not clear what the authors mean by their “prior observation.” HDX for this loop does not seem to have been referred to previously in this manuscript. Was this a previously published observation? If so, there should be a reference. The authors probably mean the β 1- β 2 loop of the blade 2 (where residue 558 is), but they should so this explicitly. The alphafold model shows two helices in this loop. Is this consistent with the absolute HDX protection values for the region corresponding to the second helix in the loop? The alphafold mode predicts that this loop interacts with the kinase domain. How do the differences in the protection factors agree with this prediction?

The prior observation that we were referring to was mentioned on lines 120-128. We observed a clipping protection when generating the full length HPK1 protein, suggesting that the protease-specific cleavage site located in the CHD was protected likely through interaction from KD with CHD.

We have now included supplemental files including uptake traces for all peptides (**Files S1 and S2**). This region shows large changes in the HDX experiment, and additionally shows some protection in the CHD construct that would be consistent with structure, see peptides 555-572, 558-564 in **File S2** (as opposed to a denatured state ensemble that exchanges at the chemical exchange limit).

Lines 290-292: The authors state: “Interestingly, in the absence of the kinase domain (CHD+4PRs), SLP76 docking still occurred but at levels comparable to Δ CHD and was still reduced relative to the full-length (Figure S7B, lower panel).

The lower figure does not seem to show the nanobit measurement for the full-length HPK1, only CHD+3PRs of CHD+4PRs. The upper panel probably has full-length kinase dead HPK1, but it means comparing one line from the lower panel with one line from the upper panel. Even if this comparison is made, it is not clear that CHD+4PRs shows reduced luminescence relative to the full-length HPK1. This is critical to the next statement “Altogether, these data suggest that presence of both the KD and CHD as well as the Y381 docking site are important for efficient SLP76 docking and optimal phosphorylation of SLP76 by the HPK1 KD.” Obviously, the kinase domain is necessary for SLP76 phosphorylation, but the observations do not show that it is important for docking.

To address the reviewer’s concern, the authors have modified **Figure S8B** to include an overlay for the plots of full-length HPK1 and Δ CHD for both plots to show the relative extent of docking for CHD+3PRs, CHD+4PRs, and other forms tested.

Lines 521-522: The kinase assay methods is confusing. It currently refers to “active mutant wild-type.” It is not clear what is “mutant wild-type.” This needs to be rephrased in a more coherent way. I am guessing that an active mutant wild-type means phosphorylation mimic mutant. This should be stated.

We have modified the nomenclature to reflect the true mutation status and also to be consistent with our prior publication from Wu et al, *Structure* (2019). The “active” mutant is now referred to as “TSEE” given the presence of Thr165Glu/Ser171Glu mutations that mimic the phosphorylated active state of the HPK1 kinase whereas the “SA” form contains a Ser171Ala mutation that mimics the inactive state.

Line 524: There is a reference to something called “Biotin-SLP76.” However, there is nothing in

the manuscript that explains what this is. Full-length SLP76 biotinylated? A SLP76 peptide? How is it biotinylated. One purpose of the methods is to facilitate repeating the work. This is not possible without explicit methods.

The Biotin-SLP76 utilized was an internally generated full-length form of SLP76 (M1-P533) expressed in *E. coli* with N-terminal His- and Avi-tags. This has been specifically added to the Materials & Methods section of the manuscript.

REVIEWERS' COMMENTS

Reviewer #1 (Remarks to the Author):

The authors have adequately addressed all my comments. I believe the manuscript is now suitable for publication in Nature Communications.

Reviewer #2 (Remarks to the Author):

The author addressed all my comments and improved the manuscript, therefore I recommend it for publication.

Reviewer #3 (Remarks to the Author):

The authors have responded adequately to all of my concerns, and I recommend that the manuscript should be accepted as it is.